# BOOTSTRAP SAMPLING RATE GREATER THAN 1.0 MAY IMPROVE RANDOM FOREST PERFORMANCE

## ABSTRACT

Random forests utilize bootstrap sampling to create an individual training set for each component tree. This involves sampling with replacement, with the number of instances equal to the size of the original training set ($N$). Research literature indicates that drawing fewer than $N$ observations can also yield satisfactory results. The ratio of the number of observations in each bootstrap sample to the total number of training instances is called the bootstrap rate (BR). Sampling more than $N$ observations (BR $> 1$) has been explored in the literature only to a limited extent and has generally proven ineffective. In this paper, we re-examine this approach using 36 diverse datasets and consider BR values ranging from 1.2 to 5.0. Contrary to previous findings, we show that such parameterization can result in statistically significant improvements in classification accuracy compared to standard settings (BR $\leq 1$). Furthermore, we investigate what the optimal BR depends on and conclude that it is more a property of the dataset than a dependence on the random forest hyperparameters. Finally, we develop a binary classifier to predict whether the optimal BR is $\leq 1$ or $> 1$ for a given dataset, achieving between 81.88% and 88.81% accuracy, depending on the experiment configuration. The code is available at: <placeholder>.

## 1 INTRODUCTION

Random forest (RF) algorithm, introduced by Breiman (2001), is an ensemble of decision trees (DTs) that collectively make decisions using either majority or soft voting. RF reduces variance, sometimes at the cost of slightly increasing bias, by introducing two sources of randomness. The first is the use of distinct subsets of features when selecting the best split at each node of the trees. The second is training each tree on a subset of observations drawn with replacement from the original training set, i.e., a bootstrap sample.

In this study, we analyze the bootstrap rate (BR), an RF hyperparameter that controls the training process and consequently affects the model's performance. BR is defined as the ratio of the number of observations in each bootstrap sample to the total number of training instances. In the literature, this parameter is also referred to as the sample rate, subsample size, bootstrap size ratio, or bag size. In his original work, Breiman (2001) used BR $= 1$. However, lower values have also been successfully applied (Martínez-Muñoz & Suárez, 2010; Adnan, 2014). When BR is low, each tree is trained on a more distinct subset of the data, which increases diversity among RF estimators. Naturally, the computational cost is reduced compared to BR $= 1$. On the other hand, the trees may become too weak, as they are trained on a relatively smaller portion of the data. For BR $= 1$, the expected fraction of unique observations from the entire dataset is 63.2%. When BR $< 1$, this fraction is even lower.

For BR $= 1$, we expect 36.8% of observations to be absent in each bootstrap sample. Intuitively, there is no obvious answer as to what would happen if BR $> 1$. A higher BR, on the one hand, causes subsets to be less diverse, but on the other hand, it includes more unique observations (i.e., more information) in each sample. We found this problem worth investigating. To our knowledge, Martínez-Muñoz & Suárez (2010) are the only ones who have analyzed BR $> 1$. However, they only considered BR $= 1.2$ and concluded that such parameterization is generally ineffective. In our work, we not only analyze BR $= 1.2$ (and lower) but also explore higher values of 2, 3, 4, and 5. Additionally, we extend the experimental setup to 18 RF configurations, compared to what appears

to be a single configuration (though this is not clearly specified) in the reference paper. Surprisingly, and in contrast to the findings of Martínez-Muñoz & Suárez (2010), we discover that BR > 1 often yields better results than conventional BR values in the range $(0, 1]$.

The primary contributions of this work can be summarized in four key points:

- To our knowledge, we are the first to analyze and shed light on what the optimal BR value depends on;
- To our knowledge, we are the first to suggest that testing BR > 1 is meaningful and often yields better results than the standard BR $\leq$ 1;
- We demonstrate that the optimal BR is only partially dependent on the RF configuration and is more a property of the dataset;
- We develop a binary classifier that, based on the class structure, predicts whether the optimal BR is $\leq$ 1 or > 1 for a given dataset, and achieves an accuracy between 81.88% and 88.81%, depending on the experimental configuration.

## 2  RELATED LITERATURE

Probst et al. (2019), in their survey on RF tuning, point out several hyperparameters that are commonly targeted by researchers when optimizing RF. The number of trees, which is the most extensively explored RF hyperparameter, was analyzed by Oshiro et al. (2012); Scornet (2017); Probst & Boulesteix (2018). The optimization of the number of attributes to consider when looking for the best split was addressed by Bernard et al. (2009); Goldstein et al. (2011). Additionally, Scornet (2017); Duroux, Roxane & Scornet, Erwan (2018) analyzed maximum tree depth.

We found the BR hyperparameter to be underresearched. Probst et al. (2019) consider it to have a minor influence on RF performance, while simultaneously stating that it is often worth tuning. Duroux, Roxane & Scornet, Erwan (2018) claim that due to the complexity of RFs, conducting a thorough theoretical analysis is challenging. As a result, most studies either overlook bootstrapping entirely (Biau et al., 2008; Ishwaran & Kogalur, 2010; Denil et al., 2013) or focus on simplified versions of RF, such as median forests (Scornet, 2017; Duroux, Roxane & Scornet, Erwan, 2018).

The study most relevant to our research was conducted by Martínez-Muñoz & Suárez (2010). First, it is the only work we found that analyzed BR > 1, although it is limited to a BR value of 1.2. Second, they examine how RF performance depends on BR. Among the 30 datasets analyzed, four types of BR curves showing the relationship between BR and classification error were identified. However, the analysis is limited to just one RF configuration and does not explore why the optimal BR may differ significantly between datasets—there is no insight provided as to why a particular curve shape is associated with a given dataset.

## 3  EXPERIMENT CONFIGURATION

Experiments were conducted on 36 diverse datasets, which underwent the following preprocessing steps: all duplicates and rows with classes occurring only once were removed. Columns with a single unique value were also dropped. Missing values in categorical features were replaced with a new category, while missing values in numerical attributes were imputed with the column mean. Finally, one-hot encoding of categorical attributes was applied before standardizing all features. Table 3 in Appendix A presents the characteristics of the datasets after the preprocessing. Our experiments include all 30 datasets used by Martínez-Muñoz & Suárez (2010) and six additional ones.

The following hyperparameters (along with BR) are considered to be the most important for RF performance (Scornet, 2017; Probst et al., 2019; Zhu et al., 2022): number of trees ($nt$); parameters controlling the size of the trees: maximum tree depth ($md$), the minimum number of instances required to split an internal node ($mn$), the minimum count of observations necessary to constitute a leaf node ($ml$); function measuring the quality of a split ($qs$); number of attributes to consider when looking for the best split ($nf$).

As the base values for these hyperparameters, we adopted the defaults from the scikit-learn 1.1.3 Python package: $nt = 100$, $md$ = None (no depth limit), $qs$ = "gini" (Gini impurity), $mn = 2$, $ml$

= 1, $nf$ = "sqrt" (square root of the number of features). We denote such a model as RF(base). Altogether, we tested RF(base) and 17 other configurations resulting from the following modifications of each single hyperparameter in RF(base):

- RF(nt_200), RF(nt_500): number of trees equals 200 or 500, respectively;

- RF(md_10), RF(md_15), RF(md_20), RF(md_25): maximum depth of a tree equals 10, 15, 20, or 25, respectively;

- RF(qs_ent): split quality is measured using Shannon entropy (information gain);

- RF(mn_3), RF(mn_4), RF(mn_6), RF(mn_8): minimum number of observations required to split an internal node is equal to 3, 4, 6, or 8, respectively;

- RF(ml_2), RF(ml_3), RF(ml_4), RF(ml_5): minimum number of instances per leaf is 2, 3, 4, or 5, respectively;

- RF(nf_log), RF(nf_all): number of features considered in a node split equals the logarithm with base 2 of the number of attributes or all features are taken into account, respectively.

The following BRs were tested: 0.2, 0.4, 0.6, 0.8, 1.0, 1.2 (as analyzed by Martínez-Muñoz & Suárez (2010)), 2.0, 3.0, 4.0, and 5.0. For each configuration, 2-fold stratified cross-validation, repeated 200 times, was applied, yielding 400 results.

## 4 RESULTS

For each dataset, we searched for the pair of RF configuration and BR that yielded the highest classification accuracy. Table 1 presents these results. Detailed results, including the mean accuracy and standard deviation for individual RF configurations and datasets, are provided in Appendix B.

**Statistical significance.** The main observation is that BR > 1 constituted the best setup in 20 out of 36 datasets. To further compare standard BRs (BR $\leq$ 1, first group) with those greater than one (second group), we performed a paired $t$-test (with the alternative hypothesis that the first sample has a greater mean than the second one) on the results of the dataset winner (best performing configuration) and results from all configurations with the other BR group. So, if the best classification accuracy was achieved by RF with BR $\leq$ 1, we compared these results with all results related to configurations with BR > 1, and vice versa. The last column of the table shows the maximum $p$-value among all $t$-tests for each dataset. We analyzed several significance levels: 0.1, 0.05, 0.01, 0.001, 0.0001, and 0.00001. Considering only conclusive results (i.e., $p$-values lower than the specified significance level), the difference in the number of datasets with the best related model with BR > 1 versus those with BR $\leq$ 1 amounted to 5, 2, -2, -4, -2, and 0, respectively. This indicates that, depending on the chosen significance level, the number of datasets with the optimal solution involving BR $\leq$ 1 is roughly comparable to those with BR > 1.

**Number of winning configurations.** Among the 18 analyzed RF configurations, only seven achieved the highest classification accuracy in at least one dataset: RF(nt_500) (20 datasets), RF(qs_ent) (5 datasets), RF(ml_5) (4 datasets), RF(mn_8) (3 datasets), RF(ml_4) (2 datasets), RF(mn_4) (1 dataset), and RF(nf_all) (1 dataset). Further analysis will concentrate on these setups.

**Frequence of winning BRs.** Fig. 1 depicts the frequency of winning BRs, both globally and for each RF parameter setting. The histogram related to RF(nt_500) is the most similar to the global one, as this model achieved the best score in 20 out of 36 datasets. For RF(ml_4) and RF(ml_5), models that restrictively control the size of the tree, BR > 1 constituted the best setup for as many as 26 datasets. It stems from the fact that, in many cases, a low number of training instances combined with a relatively high minimum number of samples required to create a leaf led to underfitted trees. Thus, high BR served as a remedy, enabling the construction of more complex models. RF(nf_all) exhibited different behavior compared to the other models. The higher the BR, the less frequently it was optimal. The key sources of diversity among the individual trees are the distinct subsets of attributes to consider when looking for the best split in each node, along with the unique bootstrap sample used in training. When all features are analyzed in a node splitting, the first source of diversity ceases to exist. Thus, to maintain an overall level of diversity, RF(nf_all) preferred lower BRs,

Table 1: Classification results. The consecutive columns present the dataset name, the optimal RF configuration, the achieved accuracy, the best BR, and the $p$-value from the conducted $t$-test.

| Dataset | Best model | Acc. [%] | Bootstrap rate | $p$-value |
|---|---|---|---|---|
| Abalone | RF(ml_5) | 26.801 | 0.2 | $< 10^{-6}$ |
| Adult | RF(ml_5) | 86.484 | 4.0 | $< 10^{-6}$ |
| Arrhythmia | RF(nf_all) | 76.161 | 1.2 | 0.305022 |
| Audiology (Standardized) | RF(mn_8) | 75.338 | 5.0 | 0.013121 |
| Australian Credit Approval | RF(nt_500) | 87.225 | 0.6 | 0.132623 |
| Balance Scale | RF(nt_500) | 85.972 | 0.2 | $< 10^{-6}$ |
| Breast Cancer Wisc. (Diag.) | RF(qs_ent) | 95.898 | 5.0 | $< 10^{-6}$ |
| Breast Cancer Wisc. (Orig.) | RF(nt_500) | 95.506 | 0.4 | 0.001910 |
| Congressional Voting Rec. | RF(mn_8) | 94.795 | 2.0 | 0.029394 |
| Echocardiogram | RF(ml_5) | 73.113 | 2.0 | 0.035933 |
| Ecoli | RF(nt_500) | 85.835 | 0.6 | 0.000097 |
| German Credit Data | RF(nt_500) | 75.467 | 1.2 | 0.079419 |
| Glass Identification | RF(qs_ent) | 75.596 | 2.0 | 0.002702 |
| Heart | RF(ml_4) | 83.324 | 0.2 | 0.000004 |
| Hepatitis | RF(nt_500) | 84.726 | 0.4 | 0.000644 |
| Horse Colic | RF(nt_500) | 86.516 | 1.0 | 0.485986 |
| Image Segmentation (Stat.) | RF(qs_ent) | 97.133 | 5.0 | $< 10^{-6}$ |
| Ionosphere | RF(nt_500) | 93.254 | 1.2 | 0.192406 |
| Iris | RF(mn_4) | 95.232 | 0.4 | 0.105220 |
| Labor Relations | RF(nt_500) | 93.608 | 1.2 | 0.004194 |
| Liver Disorders | RF(ml_4) | 59.714 | 0.2 | $< 10^{-6}$ |
| Optical Recognition (Digits) | RF(nt_500) | 97.413 | 4.0 | $< 10^{-6}$ |
| Parkinsons | RF(nt_500) | 89.306 | 5.0 | $< 10^{-6}$ |
| Pima Indians Diabetes | RF(nt_500) | 76.344 | 0.2 | 0.000018 |
| Sonar, Mines vs. Rocks | RF(qs_ent) | 81.627 | 4.0 | $< 10^{-6}$ |
| Soybean (Large) | RF(mn_8) | 92.712 | 4.0 | $< 10^{-6}$ |
| Tic-Tac-Toe Endgame | RF(nt_500) | 97.264 | 5.0 | 0.021006 |
| Thyroid Disease | RF(qs_ent) | 95.840 | 1.2 | 0.059261 |
| Vehicle Silhouettes | RF(nt_500) | 74.583 | 5.0 | 0.001911 |
| Vowel Recognition | RF(nt_500) | 92.285 | 3.0 | $< 10^{-6}$ |
| Wine | RF(nt_500) | 97.809 | 1.2 | 0.072172 |
| Ringnorm | RF(nt_500) | 92.717 | 0.6 | $< 10^{-6}$ |
| Threenorm | RF(nt_500) | 80.050 | 0.4 | 0.000654 |
| Twonorm | RF(nt_500) | 96.002 | 0.2 | $< 10^{-6}$ |
| Waveform | RF(nt_500) | 86.165 | 0.2 | $< 10^{-6}$ |
| LED Display Domain | RF(ml_5) | 66.590 | 1.0 | $< 10^{-6}$ |

which created more varied sets of samples drawn and made the trees less correlated. Looking at the BR histograms, extreme BR values (0.2 and 5.0) constituted the best solutions, both overall and for all analyzed RF configurations, the greatest number of times (the highest bar in each histogram corresponds either to 0.2 or 5.0). This suggests that the optimal BR may often be lower than 0.2 or higher than 5.0, indicating that even a broader range should be tested when tuning RF. Finally, BR = 1, defined in the original formulation of the bootstrapping procedure and the most frequently used value, performed relatively poorly. Overall, it was optimal for only two datasets. When analyzing individual RF configurations, for three of them, BR = 1 was not able to win even in one dataset. Averaging over all seven parameter settings, it was optimal for only 1.43 out of 36 datasets. Interestingly, adjacent to 1, the nonstandard BR = 1.2 was, with the exception of RF(nf_all), always better, often substantially.

**Individual dataset analysis.** Fig. 2 illustrates the relationship between the performance and BR for the analyzed RF configurations across a selected group of diverse datasets. Charts for the remaining datasets are provided in Appendix C. Our first observation is that RF(nf_all) behaves differently

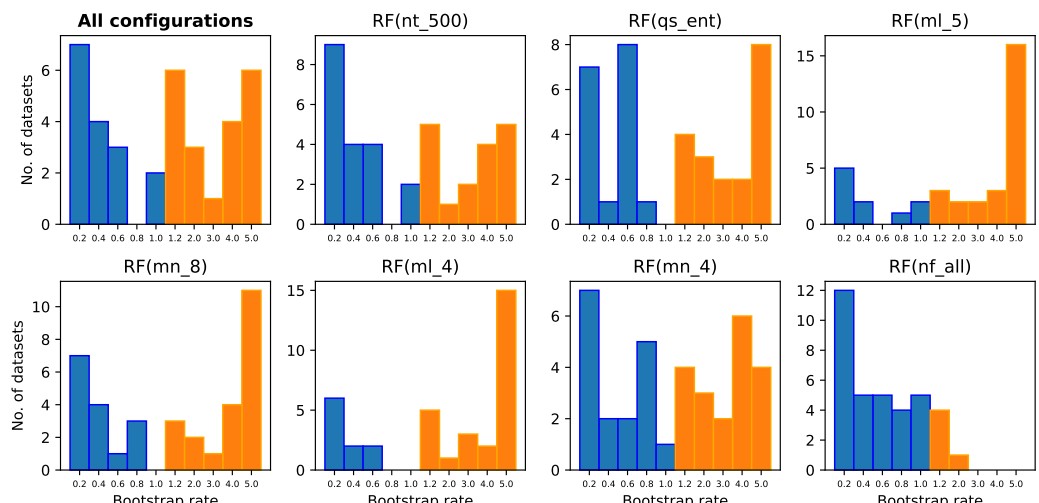

Figure 1: Distribution of winning BR across all RF configurations (top left) and among individual RF parameterizations.

from the other models. In almost all cases, it reaches optimal accuracy with a lower (or equal) BR compared to other RFs. This observation is consistent with the trends shown in Fig. 1 and the explanation provided in the previous paragraph. In most cases, the best accuracy achieved by RF(nf_all) is substantially worse than that of the other models, and after reaching the optimum, its performance declines rapidly. There are several datasets for which RF(nf_all) performs well. It achieved the best accuracy across all models on the Arrhythmia dataset and was comparable to the best on Audiology (Standardized), Tic-Tac-Toe Endgame, Pima Indians Diabetes, and Iris. The high performance of RF(nf_all) is undoubtedly related to the characteristics of the features, e.g., the abovementioned Arrhythmia has the highest number of features among all datasets. However, the relationship is more complex for the other four datasets. We hypothesize that the importance of features needs to be further assessed to gain deeper insights. Presumably, RF(nf_all) will perform well on datasets with a high proportion of insignificant or less significant features, as it may avoid building trees primarily based on these features.

**Typical BR curve shapes.**   All RF configurations other than RF(nf_all) are generally similar in terms of the characteristics of their BR curves. We identified three categories that describe how the set of curves appears:

(a) In the first and most common pattern, all curves increase to at least BR = 1.2, indicating that the optimal BR is at least 1.2. The curves then either continue to rise (usually more smoothly)/reach a plateau (first subpattern) or they oscillate/gradually decrease (second subpattern). The first subpattern can be observed in the Arrhythmia, Audiology (Standardized), Parkinsons, Breast Cancer Wisc. (Diag.), Optical Recognition (Digits), Ionosphere, Image Segmentation (Stat.), Sonar, Mines vs. Rocks, Soybean (Large), Tic-Tac-Toe Endgame, Vowel, and Recognition datasets. The second subpattern is seen in the Wine, German Credit Data, Glass Identification, Labor Relations, Thyroid Disease, Vehicle Silhouettes, and Congressional Voting Rec. datasets.

(b) In the second pattern, all curves either decrease from the very beginning (BR = 0.2) or rise to a BR in the range [0.4, 1.0] and then decline. The overall shape of the curves may be fairly smooth, as seen in the Abalone, Balance Scale, Breast Cancer Wisc. (Orig.), Heart, Liver Disorders, Twonorm, Waveform, and LED Display Domain datasets, or it may exhibit some irregularities, as observed in the Iris and Pima Indians Diabetes datasets.

(c) The third pattern is a mixture of the first and second patterns. Curves associated with some RF configurations, mainly RF(ml_4) and RF(ml_5), behave similarly to those in the first pattern, while others resemble the curves seen in the second pattern. This third pattern is present in the Adult, Aus-

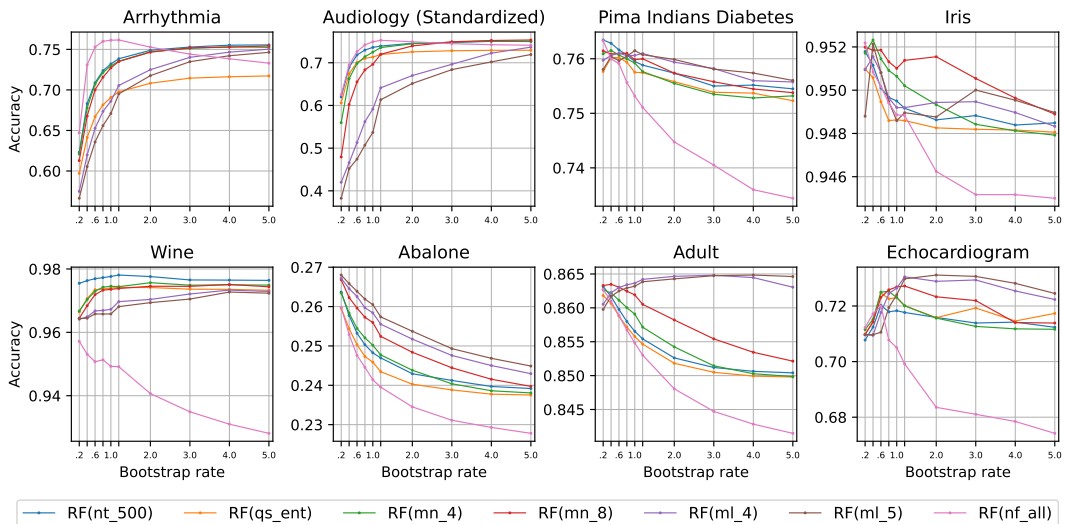

Figure 2: Characteristics of bootstrap rate curves for selected datasets.

tralian Credit Approval, Ecoli, Hepatitis, Ringnorm, Threenorm, Horse Colic, and Echocardiogram datasets. In the case of the last two, some additional irregularities in the BR curves may be observed.

The main observation from the above analysis is that BR curves associated with all RF configurations, except RF(nf_all), are fairly consistent. The first and second patterns, within which all curves exhibit similar behavior, were observed in 28 out of 36 datasets. This leads to the conclusion that the optimal BR is merely dependent on RF parameterization and is closely related to the dataset.

Naturally, the procedure for testing high BR values follows a typical 'no free lunch' scenario—while we may find RF configuration yielding better results, it comes at the cost of slower execution, as it involves sampling more observations and building trees on a larger number of instances. We did not analyze issues related to time performance, which may represent potential direction for further research.

## 5  TOWARDS UNDERSTANDING THE OPTIMAL BOOTSTRAP RATE

**Higher level approaches.**  While searching for the reasons why the BR curve differs so significantly between datasets, we began with analyzing the general properties of these datasets, such as the number of features (divided into continuous and binary) and the number of training instances. We also created new features reflecting interactions by applying arithmetic operations to the aforementioned attributes. However, neither approach helped us to understand the problem better. Next, we took a more local approach and examined whether the BR curve was associated with the number of clusters present in the data. Unfortunately, this research direction was also inconclusive. In the meantime, we observed that even small changes in the data could lead to significant changes in the shape of the BR curve and the optimal value of BR. Fig. 3 provides an example. This observation prompted us to go even more local and analyze the neighborhood of individual instances.

**Lower level approaches.**  In brief, RF is composed of DTs that cut the feature hyperspace into decision regions defined by the path leading from the root to the corresponding leaf. In a single tree, the prediction for a sample located in a particular leaf region is based on the majority class of the instances that reached that leaf during training. This means that the neighbors (specifically, their classes) of the predicted instance affect the predicted label. The same applies to RF, as it performs majority voting on predictions made by the component DTs. To analyze the structure of neighbors in each dataset, we standardized all continuous features and mapped all binary attributes to -1 and 1. We employed the Manhattan metric, which considers the distance along each feature (axis) independently, to measure the distance between observations. The Manhattan distance is generally

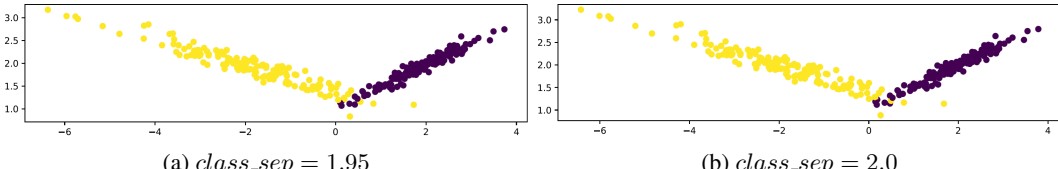

(a) $class\_sep = 1.95$                               (b) $class\_sep = 2.0$

Figure 3: An example illustrating how even small differences in the data can significantly affect the optimal BR value. Both figures (a) and (b) depict synthetically generated data using scikit-learn's $make\_classification$ method with the following parameters: $n\_samples = 300$, $n\_features = 2$, $n\_classes = 2$, $n\_clusters\_per\_class = 1$, and $random\_state = 1$. The only difference is the value of the $class\_sep$ parameter, which controls the separation between the classes. In (a), it is set to 1.95, while in (b), it is 2.0. As a result of this slight difference, the optimal BR in (a) equals 5.0, while in (b), it amounts to 0.2. All other parameters of the $make\_classification$ method remain at their default values.

a good choice in the context of a DT (and RF), which also defines a decision boundary that, at any moment, moves along only one feature.

Let us introduce the notation $k\_l$, which is the number of observations for which, among $k$ nearest neighbors, $l$ samples belong to the same class as the considered instance. Intuitively, high $k\_l$ values for low $l$, relative to $k$, indicate inhomogeneity in the data and possibly a relatively high number of outliers. For each dataset, we calculated the $k\_l$ statistics for $k \in \{1, 2, \ldots, 10\}$ and $l \in \{0, \ldots, k\}$ for each $k$. Then, we performed normalization so that for each $k$, the sum $k\_0, ..., k\_k$ equaled 100, making those values comparable between datasets of varying sizes.

For each $k\_l$ and each RF's optimal BR, including the overall best, we calculated the Spearman rank-order correlation coefficient. Table 2 presents the results for $k \in \{1, 2, \ldots, 6\}$. Our first observation is that the overall best BR is always positively correlated with $k\_k$. The highest correlation corresponds to $k = 1, 2, 3$, after which it gradually decreases. Second, for each $k\_l$ where $l \neq k$, the correlation is negative; the lower $l$ is, the stronger the correlation becomes in terms of absolute value. This means that the optimal BR for inhomogeneous datasets (high $k\_l$ values for low $l$) tends to be lower than for uniform datasets (high $k\_l$ values for $l$ close to $k$). A low BR leads to ambiguous observations being drawn less frequently. Therefore, fewer decision trees have leaves affected by such instances, and the remaining RF trees may mitigate incorrect decisions in the majority voting scheme. Conversely, for more uniform data, we hypothesize that a higher BR yields better results because it creates a bootstrap sample with more unique instances, thereby providing more information while maintaining diversity through varying the number of occurrences of these instances. Finally, an excessively high BR does not yield good results because it reduces diversity; as a consequence of the law of large numbers, the number of occurrences of individual observations converges towards each other.

Optimal BRs related to individual RF configurations share properties similar to those of the best overall BR. For all $k$, the correlation between $k\_k$ and the optimal BR is positive, gradually decreasing after $2\_2$ (or $1\_1$ in the case of RF(nf_all)). The rest of the $k\_l$ values ($l \neq k$), for $k \leq 5$, are negative, with some exceptions for $5\_4$. The trend between $k\_0$ and $k\_k$ is generally upward, but unlike the best overall BR, it is non-monotonic. RF(nf_all) exhibits this behavior as well, but the range of $k\_l$ values is visibly narrower, and some irregularity occurs for $k = 5$.

For the increasing $k$, starting from $k = 6$, the general ascending trend from $k\_0$ to $k\_k$ is maintained. However, the range $[k\_0, k\_k]$ narrows, the changes are not perfectly monotonic, and more positive values, other than $k\_k$, appear. We suppose that considering $k > 5$ becomes too general (causing the above irregularities) but still reflects the uniformity of the data (hence, an overall trend is maintained).

The absolute values of the correlation coefficients are not high. For the overall best BR, the highest Spearman rank-order correlation coefficient amounts to 0.330. This is because the function modeling the optimal BR is complex and not dependent on just one predictor. We found two ways to build attributes that are more highly correlated with the target. The first is by multiplying each feature ($k\_l$) by the number of classes in a particular dataset. Intuitively, the higher the number of classes,

Table 2: Spearman rank-order correlation coefficient between $k\_l$ and the best BR—overall (second column) and respective RF configurations (columns 3–9).

| $k\_l$ | Best RF | nt_500 | qs_ent | ml_5 | mn_8 | ml_4 | mn_4 | nf_all |
|---|---|---|---|---|---|---|---|---|
| 1_0 | -0.311 | -0.299 | -0.345 | -0.319 | -0.312 | -0.332 | -0.387 | -0.173 |
| 1_1 | 0.311 | 0.299 | 0.345 | 0.319 | 0.312 | 0.332 | 0.387 | 0.173 |
| 2_0 | -0.292 | -0.252 | -0.292 | -0.263 | -0.258 | -0.280 | -0.354 | -0.156 |
| 2_1 | -0.252 | -0.264 | -0.298 | -0.255 | -0.241 | -0.277 | -0.317 | -0.164 |
| 2_2 | 0.330 | 0.301 | 0.347 | 0.320 | 0.320 | 0.332 | 0.379 | 0.163 |
| 3_0 | -0.264 | -0.258 | -0.275 | -0.242 | -0.263 | -0.256 | -0.350 | -0.139 |
| 3_1 | -0.250 | -0.238 | -0.311 | -0.264 | -0.250 | -0.292 | -0.331 | -0.142 |
| 3_2 | -0.239 | -0.213 | -0.230 | -0.164 | -0.163 | -0.192 | -0.261 | -0.183 |
| 3_3 | 0.323 | 0.280 | 0.341 | 0.307 | 0.294 | 0.320 | 0.365 | 0.151 |
| 4_0 | -0.292 | -0.266 | -0.278 | -0.254 | -0.268 | -0.258 | -0.351 | -0.134 |
| 4_1 | -0.261 | -0.233 | -0.283 | -0.255 | -0.249 | -0.274 | -0.325 | -0.159 |
| 4_2 | -0.213 | -0.208 | -0.264 | -0.209 | -0.179 | -0.235 | -0.280 | -0.147 |
| 4_3 | -0.114 | -0.116 | -0.134 | -0.031 | -0.056 | -0.067 | -0.158 | -0.090 |
| 4_4 | 0.299 | 0.261 | 0.319 | 0.286 | 0.269 | 0.301 | 0.346 | 0.146 |
| 5_0 | -0.238 | -0.221 | -0.218 | -0.198 | -0.185 | -0.217 | -0.285 | -0.099 |
| 5_1 | -0.227 | -0.178 | -0.224 | -0.183 | -0.201 | -0.205 | -0.267 | -0.034 |
| 5_2 | -0.223 | -0.232 | -0.298 | -0.230 | -0.220 | -0.264 | -0.321 | -0.134 |
| 5_3 | -0.204 | -0.148 | -0.181 | -0.114 | -0.113 | -0.143 | -0.211 | -0.127 |
| 5_4 | -0.084 | -0.027 | -0.035 | 0.096 | 0.056 | 0.055 | -0.058 | 0.003 |
| 5_5 | 0.302 | 0.244 | 0.301 | 0.269 | 0.245 | 0.285 | 0.318 | 0.125 |
| 6_0 | -0.213 | -0.186 | -0.170 | -0.165 | -0.149 | -0.182 | -0.251 | -0.080 |
| 6_1 | -0.156 | -0.134 | -0.183 | -0.127 | -0.165 | -0.156 | -0.223 | 0.031 |
| 6_2 | -0.234 | -0.239 | -0.292 | -0.225 | -0.214 | -0.261 | -0.328 | -0.154 |
| 6_3 | -0.158 | -0.128 | -0.199 | -0.167 | -0.136 | -0.198 | -0.213 | -0.015 |
| 6_4 | -0.109 | -0.041 | -0.062 | 0.039 | 0.039 | 0.008 | -0.086 | -0.083 |
| 6_5 | -0.013 | 0.030 | -0.001 | 0.169 | 0.125 | 0.129 | 0.030 | 0.080 |
| 6_6 | 0.265 | 0.204 | 0.261 | 0.220 | 0.190 | 0.235 | 0.260 | 0.080 |

the lower the probability that outliers (ambiguous observations) from the same class, potentially forming a decision leaf in a tree, are drawn. Indeed, the correlation coefficients move upward. The negative ones become closer to zero or even turn positive. For example, for the overall best BR, 2_0, 3_2, and 4_3 increase from -0.292, -0.239, and -0.114 to -0.191, -0.041, and 0.038, respectively. Similarly, the positive values rise even higher. For instance, 2_2, 3_3, and 4_4 increase from 0.330, 0.323, and 0.299 to 0.456, 0.493, and 0.454, respectively.

The second way to create predictors with a higher correlation to the target is to introduce features that represent interactions between existing attributes $k\_l$, where $k \in \{1, 2, \ldots, 10\}$ and $l \in \{0, \ldots, k\}$ for each $k$. More specifically, for all pairs of distinct features, we perform division, subtraction, multiplication, and addition (the first two in both directions). Additionally, each attribute is multiplied by itself and added to itself, creating two additional features. In this way, 12 620 new attributes are created. The features most positively correlated with the target are 9_2/2_0 (9_2 divided by 2_0), 10_2/3_0, and 10_2/4_0. The respective correlation coefficients are 0.607, 0.595, and 0.595.

**Bootstrap rate prediction.** To further assess how well the above set of attributes describes the problem, we used them (along with base $k\_l$ statistics) to build a binary classifier predicting whether the optimal BR across all RF configurations is $\leq 1$ or $> 1$ for a given dataset. As in the main experiments, we tested all 18 RF configurations and 10 BR values. Due to the limited number of observations—each corresponding to one of 36 datasets—we performed Leave-Two-Out Cross-Validation in all possible variants, yielding 320 train-validation splits. Given the high dimensionality

(12 685 features), the initial results were poor. To address this, we reduced the number of input features to the $k$ highest correlated with the target, with $k$ ranging from 1 to 10, based on the absolute value of the Spearman rank-order coefficient, calculated separately for each run on the training instances. The best classification accuracy, averaged over all 320 runs, equaled 81.88% and was achieved by RF(nt_200) with BR = 0.4, using seven attributes.

Differences in classification accuracies between different BRs are sometimes marginal. Therefore, another experiment was conducted, this time focusing only on observations with undisputed labels. We assumed these are datasets for which the corresponding $p$-value in Table 1 is at most 0.01. A total of 24 observations met this condition. The remaining experimental configuration was the same as in the previous experiment. Leave-Two-Out Cross-Validation was performed on all possible 143 train-validation splits. The highest accuracy, 88.81%, was achieved by RF(nf_all) with BR = 0.8, using only three features.

In both of the above experiments, the number of training instances was low: 36 and 24, respectively. We believe that a simple increase in these numbers may lead to further improvements in performance. Additionally, both datasets were well-balanced, with the majority class constituting 55.56% and 54.17%, respectively. Thus, we conclude that the proposed attributes, which enabled us to achieve accuracies of 81.88% and 88.81%, can be considered as effective descriptors of the analyzed problem.

## 6 CONCLUSIONS AND FUTURE WORK

In this paper, we analyze the BR hyperparameter in RF. To the best of our knowledge, this work is the first to shed light on what the optimal BR value depends on and to demonstrate that it is often greater than 1, thus exceeds the standard values within the $(0, 1]$ range. We also show that the optimal BR value is largely independent of the other hyperparameters of the RF. In fact, most RF configurations are highly correlated in terms of the BR curve, which makes the optimal BR value more of a property of the dataset.

Our main conclusion stating that BR > 1 often yields superior results and is worth considering contradicts the findings of the baseline reference paper (Martínez-Muñoz & Suárez, 2010). We identify two main reasons for this. First, the authors of (Martínez-Muñoz & Suárez, 2010) stopped their analysis at BR = 1.2 and did not explore higher values. Second, they most likely tested only one RF configuration. In fact, they did not provide any details regarding the RF hyperparameters, other than stating that the ensemble was composed of 200 unpruned CART trees (Breiman et al., 1984).

Prediction of the optimal BR value is a complex task, highly dependent on the local class structure. In this work, we propose to calculate $k\_l$ statistics, which reflect the number of observations from the same class as the instances considered in their neighborhood and use them to calculate the correlations with the optimal BR value. While this approach works generally well, we believe that it cas still be improved through describing the local class structure in a different, possibly more precise way.

Considering nearest neighbors assumes that the analyzed observation is located at the center of the decision regions and that these regions form a hypercube, meaning all decision hyperplanes are equally distant from the analyzed sample. This is a simplification, as in real-world scenarios, the range of features corresponding to a decision subspace is usually unequal, and no point lies exactly at the center of all feature ranges. Therefore, analyzing an instance's neighborhood by sampling each feature range to reflect the different decision subspaces to which a particular instance may belong may be a viable approach.

Another research direction worth exploring is to extend $k\_l$ to more detailed statistics that specify the number of neighbors from each individual class. $k\_l$ can be interpreted as the number of instances for which, among the $k$ nearest neighbors, $k - l$ observations belong to classes other than the one under consideration. We believe that the distribution of these classes may be useful in predicting the optimal BR. The more uniform the distribution (with no dominant class), the easier it is for an ambiguous example to outvote the correct class in a majority or soft voting scheme.

Finally, we examined three well-established ML libraries: scikit-learn, Weka, and H2O.ai. In all of them, values of the BR hyperparameter greater than one are disabled in their RF implementations. Based on our findings, we recommend that the developers of ML libraries consider making this feature available.

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

# Appendices

## A DATASETS

Table 3: Dataset characteristics. The subsequent columns refer to the dataset name, the number of numerical and binary features, the number of observations, and the count of classes. The first 31 datasets presented in the table come from the UCI Machine Learning Repository (Kelly et al., 2023). The next four are from Breiman (1998), and the last one is from Breiman et al. (1984).

| Dataset | Numerical features | Binary features | Observations | Classes |
|---|---|---|---|---|
| Abalone | 7 | 3 | 4172 | 23 |
| Adult | 6 | 85 | 48790 | 2 |
| Arrhythmia | 194 | 64 | 420 | 12 |
| Audiology (Standardized) | 0 | 89 | 171 | 18 |
| Australian Credit Approval | 6 | 32 | 690 | 2 |
| Balance Scale | 0 | 20 | 625 | 3 |
| Breast Cancer Wisc. (Diag.) | 30 | 0 | 569 | 2 |
| Breast Cancer Wisc. (Orig.) | 9 | 0 | 449 | 2 |
| Congressional Voting Rec. | 0 | 48 | 342 | 2 |
| Echocardiogram | 6 | 1 | 62 | 2 |
| Ecoli | 5 | 1 | 336 | 8 |
| German Credit Data | 6 | 53 | 1000 | 2 |
| Glass Identification | 9 | 0 | 213 | 6 |
| Heart | 7 | 13 | 270 | 2 |
| Hepatitis | 6 | 27 | 148 | 2 |
| Horse Colic | 7 | 140 | 368 | 2 |
| Image Segmentation (Stat.) | 18 | 0 | 2086 | 7 |
| Ionosphere | 32 | 1 | 350 | 2 |
| Iris | 4 | 0 | 149 | 3 |
| Labor Relations | 8 | 29 | 57 | 2 |
| Liver Disorders | 5 | 0 | 341 | 2 |
| Optical Recognition (Digits) | 61 | 0 | 1797 | 10 |
| Parkinsons | 22 | 0 | 195 | 2 |
| Pima Indians Diabetes | 8 | 0 | 768 | 2 |
| Sonar, Mines vs. Rocks | 60 | 0 | 208 | 2 |
| Soybean (Large) | 0 | 132 | 631 | 19 |
| Tic-Tac-Toe Endgame | 0 | 27 | 958 | 2 |
| Thyroid Disease | 5 | 0 | 215 | 3 |
| Vehicle Silhouettes | 18 | 0 | 845 | 4 |
| Vowel Recognition | 10 | 0 | 990 | 11 |
| Wine | 13 | 0 | 178 | 3 |
| Ringnorm | 20 | 0 | 300 | 2 |
| Threenorm | 20 | 0 | 300 | 2 |
| Twonorm | 20 | 0 | 300 | 2 |
| Waveform | 21 | 0 | 300 | 3 |
| LED Display Domain | 0 | 24 | 200 | 10 |

# B  DETAILED RESULTS

Table 4: Classification accuracy (mean ± standard deviation) for the Abalone dataset.

| BR | RF(nt_500) | RF(qs_ent) | RF(mn_4) | RF(mn_8) | RF(ml_4) | RF(ml_5) | RF(nf_all) |
|---|---|---|---|---|---|---|---|
| 0.2 | 26.368 ± 0.719 | 25.959 ± 0.753 | 26.345 ± 0.759 | 26.682 ± 0.766 | 26.718 ± 0.711 | 26.801 ± 0.703 | 25.984 ± 0.799 |
| 0.4 | 25.777 ± 0.744 | 25.439 ± 0.769 | 25.835 ± 0.764 | 26.229 ± 0.754 | 26.457 ± 0.746 | 26.584 ± 0.714 | 25.290 ± 0.766 |
| 0.6 | 25.323 ± 0.741 | 25.032 ± 0.750 | 25.447 ± 0.748 | 25.963 ± 0.774 | 26.256 ± 0.750 | 26.388 ± 0.740 | 24.761 ± 0.781 |
| 0.8 | 25.031 ± 0.746 | 24.732 ± 0.780 | 25.206 ± 0.767 | 25.729 ± 0.747 | 25.970 ± 0.775 | 26.190 ± 0.803 | 24.458 ± 0.777 |
| 1.0 | 24.825 ± 0.750 | 24.588 ± 0.719 | 25.017 ± 0.762 | 25.595 ± 0.762 | 25.840 ± 0.775 | 26.050 ± 0.730 | 24.142 ± 0.731 |
| 1.2 | 24.696 ± 0.726 | 24.346 ± 0.720 | 24.773 ± 0.721 | 25.243 ± 0.761 | 25.553 ± 0.735 | 25.738 ± 0.770 | 23.953 ± 0.724 |
| 2.0 | 24.294 ± 0.752 | 24.027 ± 0.726 | 24.383 ± 0.751 | 24.840 ± 0.768 | 25.173 ± 0.771 | 25.373 ± 0.797 | 23.452 ± 0.776 |
| 3.0 | 24.123 ± 0.747 | 23.883 ± 0.739 | 24.038 ± 0.776 | 24.445 ± 0.773 | 24.758 ± 0.739 | 24.932 ± 0.753 | 23.112 ± 0.814 |
| 4.0 | 23.970 ± 0.710 | 23.775 ± 0.714 | 23.861 ± 0.740 | 24.154 ± 0.738 | 24.502 ± 0.772 | 24.686 ± 0.753 | 22.930 ± 0.787 |
| 5.0 | 23.918 ± 0.725 | 23.753 ± 0.760 | 23.806 ± 0.723 | 23.972 ± 0.733 | 24.295 ± 0.713 | 24.489 ± 0.800 | 22.779 ± 0.835 |

Table 5: Results for the Adult dataset.

| BR | RF(nt_500) | RF(qs_ent) | RF(mn_4) | RF(mn_8) | RF(ml_4) | RF(ml_5) | RF(nf_all) |
|---|---|---|---|---|---|---|---|
| 0.2 | 86.315 ± 0.151 | 86.182 ± 0.152 | 86.266 ± 0.146 | 86.329 ± 0.152 | 86.053 ± 0.160 | 85.977 ± 0.157 | 86.270 ± 0.150 |
| 0.4 | 86.168 ± 0.151 | 86.068 ± 0.154 | 86.220 ± 0.144 | 86.347 ± 0.146 | 86.224 ± 0.155 | 86.167 ± 0.157 | 86.098 ± 0.154 |
| 0.6 | 85.977 ± 0.153 | 85.879 ± 0.157 | 86.112 ± 0.152 | 86.305 ± 0.146 | 86.300 ± 0.154 | 86.250 ± 0.152 | 85.886 ± 0.160 |
| 0.8 | 85.798 ± 0.159 | 85.717 ± 0.165 | 86.002 ± 0.158 | 86.245 ± 0.157 | 86.341 ± 0.154 | 86.292 ± 0.153 | 85.673 ± 0.168 |
| 1.0 | 85.654 ± 0.156 | 85.576 ± 0.157 | 85.908 ± 0.150 | 86.192 ± 0.150 | 86.363 ± 0.151 | 86.319 ± 0.155 | 85.482 ± 0.171 |
| 1.2 | 85.542 ± 0.157 | 85.464 ± 0.155 | 85.715 ± 0.158 | 86.052 ± 0.152 | 86.420 ± 0.152 | 86.385 ± 0.155 | 85.302 ± 0.171 |
| 2.0 | 85.261 ± 0.157 | 85.182 ± 0.158 | 85.424 ± 0.156 | 85.822 ± 0.153 | 86.465 ± 0.153 | 86.428 ± 0.157 | 84.805 ± 0.174 |
| 3.0 | 85.120 ± 0.154 | 85.049 ± 0.158 | 85.145 ± 0.157 | 85.542 ± 0.160 | 86.481 ± 0.153 | 86.475 ± 0.156 | 84.471 ± 0.192 |
| 4.0 | 85.063 ± 0.157 | 84.993 ± 0.157 | 85.028 ± 0.162 | 85.344 ± 0.154 | 86.447 ± 0.156 | 86.484 ± 0.155 | 84.284 ± 0.202 |
| 5.0 | 85.041 ± 0.157 | 84.979 ± 0.163 | 84.986 ± 0.163 | 85.214 ± 0.159 | 86.305 ± 0.156 | 86.462 ± 0.151 | 84.148 ± 0.211 |

Table 6: Results for the Arrhythmia dataset.

| BR | RF(nt_500) | RF(qs_ent) | RF(mn_4) | RF(mn_8) | RF(ml_4) | RF(ml_5) | RF(nf_all) |
|---|---|---|---|---|---|---|---|
| 0.2 | 62.300 ± 1.482 | 59.704 ± 1.396 | 62.098 ± 1.578 | 61.287 ± 1.432 | 57.487 ± 0.825 | 56.637 ± 0.439 | 64.712 ± 1.938 |
| 0.4 | 68.310 ± 1.629 | 64.137 ± 1.667 | 67.794 ± 1.825 | 66.781 ± 1.821 | 61.971 ± 1.519 | 60.544 ± 1.315 | 73.096 ± 2.006 |
| 0.6 | 70.875 ± 1.652 | 66.696 ± 1.631 | 70.719 ± 1.837 | 70.010 ± 1.815 | 65.270 ± 1.722 | 63.580 ± 1.615 | 75.290 ± 1.896 |
| 0.8 | 72.371 ± 1.683 | 68.155 ± 1.724 | 72.150 ± 1.771 | 71.561 ± 1.862 | 67.346 ± 1.845 | 65.598 ± 1.804 | 75.999 ± 1.944 |
| 1.0 | 73.208 ± 1.767 | 69.061 ± 1.656 | 72.975 ± 1.874 | 72.714 ± 1.928 | 68.610 ± 1.788 | 67.092 ± 1.843 | 76.130 ± 1.929 |
| 1.2 | 73.844 ± 1.792 | 69.758 ± 1.631 | 73.538 ± 1.945 | 73.505 ± 1.995 | 70.548 ± 1.793 | 69.507 ± 1.780 | 76.161 ± 1.949 |
| 2.0 | 74.887 ± 1.863 | 70.823 ± 1.605 | 74.706 ± 1.863 | 74.654 ± 1.939 | 72.496 ± 1.762 | 71.760 ± 1.868 | 75.277 ± 2.320 |
| 3.0 | 75.289 ± 1.910 | 71.452 ± 1.740 | 75.127 ± 1.972 | 75.193 ± 2.043 | 74.018 ± 1.969 | 73.482 ± 1.882 | 74.417 ± 2.534 |
| 4.0 | 75.518 ± 1.941 | 71.625 ± 1.742 | 75.327 ± 1.915 | 75.271 ± 1.865 | 74.660 ± 1.920 | 74.217 ± 1.858 | 73.840 ± 2.693 |
| 5.0 | 75.554 ± 1.831 | 71.726 ± 1.644 | 75.250 ± 1.948 | 75.427 ± 1.892 | 75.024 ± 1.807 | 74.637 ± 1.956 | 73.299 ± 2.860 |

Table 7: Results for the Audiology (Standardized) dataset.

| BR | RF(nt_500) | RF(qs_ent) | RF(mn_4) | RF(mn_8) | RF(ml_4) | RF(ml_5) | RF(nf_all) |
|---|---|---|---|---|---|---|---|
| 0.2 | 62.012 ± 3.483 | 60.583 ± 3.708 | 55.974 ± 3.811 | 47.936 ± 2.531 | 41.996 ± 4.582 | 38.258 ± 6.699 | 62.505 ± 3.902 |
| 0.4 | 68.927 ± 3.394 | 67.429 ± 3.510 | 66.274 ± 3.570 | 60.223 ± 3.823 | 46.573 ± 2.348 | 45.205 ± 2.154 | 69.445 ± 3.565 |
| 0.6 | 71.831 ± 3.390 | 70.062 ± 3.607 | 69.844 ± 3.525 | 65.517 ± 3.301 | 51.308 ± 3.124 | 47.432 ± 2.393 | 72.611 ± 3.894 |
| 0.8 | 72.977 ± 3.460 | 71.022 ± 3.488 | 71.411 ± 3.601 | 68.334 ± 3.502 | 56.200 ± 3.591 | 50.707 ± 3.160 | 74.181 ± 4.129 |
| 1.0 | 73.600 ± 3.529 | 71.338 ± 3.652 | 72.482 ± 3.604 | 69.635 ± 3.662 | 59.150 ± 3.630 | 53.672 ± 3.550 | 74.912 ± 4.221 |
| 1.2 | 73.914 ± 3.642 | 71.940 ± 3.598 | 73.553 ± 3.628 | 71.984 ± 3.495 | 64.130 ± 3.055 | 61.341 ± 3.446 | 75.257 ± 4.040 |
| 2.0 | 74.563 ± 3.634 | 72.556 ± 3.819 | 74.459 ± 3.808 | 73.940 ± 3.741 | 67.001 ± 2.997 | 65.144 ± 2.945 | 74.918 ± 4.231 |
| 3.0 | 74.742 ± 3.833 | 72.814 ± 3.668 | 74.812 ± 3.897 | 74.913 ± 3.833 | 69.641 ± 3.490 | 68.392 ± 3.249 | 74.473 ± 4.305 |
| 4.0 | 75.017 ± 3.864 | 72.908 ± 3.733 | 75.122 ± 3.949 | 75.216 ± 3.679 | 72.248 ± 3.682 | 70.200 ± 3.341 | 74.227 ± 4.441 |
| 5.0 | 74.980 ± 3.860 | 72.929 ± 3.729 | 75.037 ± 3.703 | 75.338 ± 3.621 | 73.653 ± 3.646 | 71.930 ± 3.452 | 74.101 ± 4.527 |

Table 8: Results for the Australian Credit Approval dataset.

| BR | RF(nt_500) | RF(qs_ent) | RF(mn_4) | RF(mn_8) | RF(ml_4) | RF(ml_5) | RF(nf_all) |
|---|---|---|---|---|---|---|---|
| 0.2 | 86.946 ± 1.397 | 86.680 ± 1.431 | 86.727 ± 1.451 | 86.634 ± 1.430 | 86.312 ± 1.469 | 86.151 ± 1.512 | 86.208 ± 1.450 |
| 0.4 | 87.151 ± 1.370 | 86.986 ± 1.429 | 86.974 ± 1.479 | 86.810 ± 1.413 | 86.714 ± 1.419 | 86.564 ± 1.400 | 86.525 ± 1.461 |
| 0.6 | 87.225 ± 1.399 | 87.087 ± 1.415 | 87.062 ± 1.422 | 86.945 ± 1.355 | 86.920 ± 1.407 | 86.773 ± 1.429 | 86.541 ± 1.443 |
| 0.8 | 87.199 ± 1.411 | 87.098 ± 1.436 | 87.135 ± 1.393 | 87.067 ± 1.400 | 86.947 ± 1.409 | 86.812 ± 1.436 | 86.404 ± 1.490 |
| 1.0 | 87.214 ± 1.409 | 87.070 ± 1.430 | 87.089 ± 1.368 | 87.062 ± 1.376 | 86.953 ± 1.409 | 86.901 ± 1.454 | 86.130 ± 1.484 |
| 1.2 | 87.188 ± 1.367 | 87.114 ± 1.363 | 87.059 ± 1.393 | 87.055 ± 1.363 | 86.990 ± 1.430 | 86.956 ± 1.371 | 85.850 ± 1.602 |
| 2.0 | 87.051 ± 1.356 | 87.018 ± 1.413 | 87.022 ± 1.401 | 87.025 ± 1.412 | 87.105 ± 1.459 | 87.018 ± 1.405 | 84.980 ± 1.749 |
| 3.0 | 86.989 ± 1.373 | 86.878 ± 1.381 | 86.917 ± 1.342 | 86.999 ± 1.368 | 87.096 ± 1.366 | 87.104 ± 1.425 | 84.350 ± 1.891 |
| 4.0 | 86.944 ± 1.346 | 86.821 ± 1.385 | 86.841 ± 1.417 | 86.888 ± 1.367 | 87.078 ± 1.373 | 87.070 ± 1.410 | 83.967 ± 1.852 |
| 5.0 | 86.911 ± 1.375 | 86.824 ± 1.366 | 86.844 ± 1.423 | 86.891 ± 1.388 | 87.073 ± 1.413 | 87.076 ± 1.345 | 83.723 ± 1.931 |

Table 9: Results for the Balance Scale dataset.

| BR | RF(nt_500) | RF(qs_ent) | RF(mn_4) | RF(mn_8) | RF(ml_4) | RF(ml_5) | RF(nf_all) |
|---|---|---|---|---|---|---|---|
| 0.2 | 85.972 ± 1.474 | 85.018 ± 1.595 | 85.404 ± 1.599 | 85.519 ± 1.766 | 85.454 ± 1.914 | 85.296 ± 2.031 | 84.374 ± 1.747 |
| 0.4 | 84.732 ± 1.452 | 84.095 ± 1.441 | 84.649 ± 1.527 | 84.895 ± 1.719 | 85.386 ± 1.800 | 85.283 ± 1.877 | 82.641 ± 1.830 |
| 0.6 | 83.947 ± 1.427 | 83.416 ± 1.440 | 84.227 ± 1.549 | 84.487 ± 1.652 | 84.906 ± 1.668 | 84.959 ± 1.792 | 81.262 ± 1.775 |
| 0.8 | 83.329 ± 1.493 | 82.786 ± 1.435 | 83.811 ± 1.464 | 84.268 ± 1.595 | 84.635 ± 1.705 | 84.664 ± 1.765 | 80.093 ± 1.887 |
| 1.0 | 82.831 ± 1.449 | 82.333 ± 1.582 | 83.593 ± 1.489 | 84.180 ± 1.585 | 84.490 ± 1.682 | 84.547 ± 1.719 | 79.090 ± 1.913 |
| 1.2 | 82.483 ± 1.445 | 81.907 ± 1.519 | 82.909 ± 1.497 | 83.936 ± 1.582 | 84.319 ± 1.667 | 84.341 ± 1.708 | 78.248 ± 1.864 |
| 2.0 | 81.581 ± 1.538 | 81.143 ± 1.593 | 81.958 ± 1.603 | 83.419 ± 1.518 | 83.913 ± 1.538 | 84.051 ± 1.590 | 76.734 ± 1.969 |
| 3.0 | 81.256 ± 1.579 | 80.743 ± 1.629 | 81.097 ± 1.655 | 82.447 ± 1.562 | 83.619 ± 1.497 | 83.661 ± 1.561 | 76.189 ± 2.062 |
| 4.0 | 81.029 ± 1.563 | 80.670 ± 1.628 | 80.938 ± 1.604 | 81.811 ± 1.567 | 83.263 ± 1.442 | 83.512 ± 1.522 | 75.956 ± 2.133 |
| 5.0 | 81.015 ± 1.614 | 80.616 ± 1.625 | 80.712 ± 1.610 | 81.326 ± 1.583 | 82.771 ± 1.516 | 83.185 ± 1.564 | 75.788 ± 2.113 |

Table 10: Results for the Breast Cancer Wisc. (Diag.) dataset.

| BR | RF(nt_500) | RF(qs_ent) | RF(mn_4) | RF(mn_8) | RF(ml_4) | RF(ml_5) | RF(nf_all) |
|---|---|---|---|---|---|---|---|
| 0.2 | 94.673 ± 1.126 | 94.598 ± 1.201 | 94.460 ± 1.194 | 94.289 ± 1.163 | 93.794 ± 1.273 | 93.567 ± 1.316 | 94.367 ± 1.365 |
| 0.4 | 95.076 ± 1.081 | 95.080 ± 1.171 | 94.880 ± 1.112 | 94.664 ± 1.121 | 94.487 ± 1.222 | 94.291 ± 1.277 | 94.704 ± 1.257 |
| 0.6 | 95.257 ± 1.111 | 95.314 ± 1.163 | 95.147 ± 1.130 | 94.924 ± 1.117 | 94.730 ± 1.144 | 94.615 ± 1.193 | 94.876 ± 1.313 |
| 0.8 | 95.360 ± 1.153 | 95.494 ± 1.144 | 95.241 ± 1.203 | 95.061 ± 1.187 | 94.922 ± 1.164 | 94.746 ± 1.194 | 94.942 ± 1.350 |
| 1.0 | 95.459 ± 1.138 | 95.606 ± 1.171 | 95.346 ± 1.140 | 95.122 ± 1.163 | 94.970 ± 1.146 | 94.829 ± 1.206 | 94.930 ± 1.383 |
| 1.2 | 95.540 ± 1.165 | 95.675 ± 1.168 | 95.511 ± 1.186 | 95.309 ± 1.152 | 95.176 ± 1.188 | 95.087 ± 1.168 | 94.822 ± 1.371 |
| 2.0 | 95.651 ± 1.202 | 95.870 ± 1.110 | 95.710 ± 1.165 | 95.600 ± 1.181 | 95.387 ± 1.183 | 95.265 ± 1.172 | 94.536 ± 1.374 |
| 3.0 | 95.724 ± 1.168 | 95.871 ± 1.113 | 95.743 ± 1.188 | 95.642 ± 1.179 | 95.544 ± 1.167 | 95.500 ± 1.163 | 94.256 ± 1.398 |
| 4.0 | 95.692 ± 1.172 | 95.897 ± 1.108 | 95.714 ± 1.188 | 95.699 ± 1.186 | 95.607 ± 1.156 | 95.518 ± 1.146 | 94.154 ± 1.462 |
| 5.0 | 95.691 ± 1.152 | 95.898 ± 1.095 | 95.702 ± 1.178 | 95.704 ± 1.151 | 95.619 ± 1.166 | 95.550 ± 1.176 | 94.007 ± 1.442 |

Table 11: Results for the Breast Cancer Wisc. (Orig.) dataset.

| BR | RF(nt_500) | RF(qs_ent) | RF(mn_4) | RF(mn_8) | RF(ml_4) | RF(ml_5) | RF(nf_all) |
|---|---|---|---|---|---|---|---|
| 0.2 | 95.366 ± 1.105 | 95.212 ± 1.144 | 95.321 ± 1.128 | 95.454 ± 1.123 | 95.458 ± 1.128 | 95.423 ± 1.112 | 94.517 ± 1.298 |
| 0.4 | 95.506 ± 1.085 | 95.309 ± 1.113 | 95.350 ± 1.113 | 95.474 ± 1.142 | 95.422 ± 1.160 | 95.386 ± 1.200 | 94.552 ± 1.255 |
| 0.6 | 95.491 ± 1.085 | 95.362 ± 1.127 | 95.440 ± 1.112 | 95.501 ± 1.118 | 95.435 ± 1.138 | 95.415 ± 1.170 | 94.570 ± 1.286 |
| 0.8 | 95.457 ± 1.070 | 95.342 ± 1.135 | 95.403 ± 1.103 | 95.440 ± 1.122 | 95.411 ± 1.124 | 95.376 ± 1.133 | 94.521 ± 1.274 |
| 1.0 | 95.429 ± 1.055 | 95.287 ± 1.087 | 95.355 ± 1.097 | 95.433 ± 1.116 | 95.401 ± 1.099 | 95.382 ± 1.105 | 94.407 ± 1.309 |
| 1.2 | 95.378 ± 1.077 | 95.247 ± 1.081 | 95.312 ± 1.109 | 95.394 ± 1.120 | 95.393 ± 1.130 | 95.403 ± 1.118 | 94.336 ± 1.357 |
| 2.0 | 95.262 ± 1.086 | 95.162 ± 1.109 | 95.176 ± 1.112 | 95.233 ± 1.095 | 95.341 ± 1.111 | 95.324 ± 1.108 | 93.919 ± 1.473 |
| 3.0 | 95.166 ± 1.105 | 95.133 ± 1.110 | 95.091 ± 1.124 | 95.151 ± 1.142 | 95.249 ± 1.138 | 95.279 ± 1.174 | 93.516 ± 1.569 |
| 4.0 | 95.155 ± 1.134 | 95.102 ± 1.124 | 95.026 ± 1.140 | 95.089 ± 1.124 | 95.179 ± 1.132 | 95.207 ± 1.145 | 93.316 ± 1.590 |
| 5.0 | 95.115 ± 1.120 | 95.064 ± 1.128 | 95.024 ± 1.153 | 95.047 ± 1.133 | 95.145 ± 1.104 | 95.183 ± 1.107 | 93.100 ± 1.618 |

Table 12: Results for the Congressional Voting Rec. dataset.

| BR | RF(nt_500) | RF(qs_ent) | RF(mn_4) | RF(mn_8) | RF(ml_4) | RF(ml_5) | RF(nf_all) |
|---|---|---|---|---|---|---|---|
| 0.2 | 93.690 ± 1.487 | 93.918 ± 1.474 | 93.819 ± 1.437 | 93.687 ± 1.439 | 92.997 ± 1.550 | 92.795 ± 1.658 | 94.404 ± 1.337 |
| 0.4 | 94.355 ± 1.520 | 94.412 ± 1.509 | 94.278 ± 1.471 | 93.974 ± 1.410 | 93.683 ± 1.421 | 93.534 ± 1.469 | 94.373 ± 1.394 |
| 0.6 | 94.592 ± 1.480 | 94.640 ± 1.489 | 94.509 ± 1.517 | 94.202 ± 1.429 | 93.807 ± 1.380 | 93.747 ± 1.428 | 94.301 ± 1.448 |
| 0.8 | 94.661 ± 1.494 | 94.692 ± 1.471 | 94.637 ± 1.519 | 94.377 ± 1.483 | 93.886 ± 1.462 | 93.800 ± 1.449 | 94.156 ± 1.453 |
| 1.0 | 94.702 ± 1.431 | 94.683 ± 1.451 | 94.686 ± 1.490 | 94.512 ± 1.500 | 94.028 ± 1.489 | 93.890 ± 1.455 | 93.915 ± 1.579 |
| 1.2 | 94.681 ± 1.457 | 94.662 ± 1.442 | 94.763 ± 1.480 | 94.652 ± 1.504 | 94.276 ± 1.524 | 94.099 ± 1.484 | 93.702 ± 1.684 |
| 2.0 | 94.675 ± 1.426 | 94.604 ± 1.443 | 94.753 ± 1.455 | 94.795 ± 1.474 | 94.481 ± 1.540 | 94.317 ± 1.505 | 93.139 ± 1.782 |
| 3.0 | 94.620 ± 1.413 | 94.569 ± 1.389 | 94.605 ± 1.457 | 94.760 ± 1.461 | 94.705 ± 1.485 | 94.620 ± 1.515 | 92.788 ± 1.878 |
| 4.0 | 94.615 ± 1.413 | 94.539 ± 1.423 | 94.575 ± 1.394 | 94.703 ± 1.433 | 94.756 ± 1.501 | 94.722 ± 1.513 | 92.721 ± 1.931 |
| 5.0 | 94.621 ± 1.432 | 94.520 ± 1.451 | 94.520 ± 1.404 | 94.633 ± 1.425 | 94.791 ± 1.472 | 94.769 ± 1.483 | 92.645 ± 1.922 |

Table 13: Results for the Echocardiogram dataset.

| BR | RF(nt_500) | RF(qs_ent) | RF(mn_4) | RF(mn_8) | RF(ml_4) | RF(ml_5) | RF(nf_all) |
|---|---|---|---|---|---|---|---|
| 0.2 | 70.774 ± 2.633 | 70.992 ± 2.462 | 71.153 ± 2.174 | 70.968 ± 0.000 | 70.968 ± 0.000 | 70.968 ± 0.000 | 71.226 ± 3.311 |
| 0.4 | 71.242 ± 4.116 | 71.435 ± 3.788 | 71.524 ± 3.908 | 71.427 ± 3.500 | 70.944 ± 0.665 | 70.968 ± 0.000 | 71.718 ± 4.577 |
| 0.6 | 72.024 ± 4.493 | 72.468 ± 4.497 | 72.500 ± 4.746 | 72.315 ± 4.755 | 71.766 ± 3.347 | 71.056 ± 2.374 | 72.000 ± 5.264 |
| 0.8 | 71.790 ± 5.039 | 72.258 ± 5.018 | 72.524 ± 4.697 | 72.581 ± 4.823 | 72.444 ± 3.855 | 71.944 ± 3.446 | 70.774 ± 6.117 |
| 1.0 | 71.831 ± 4.925 | 72.298 ± 5.077 | 72.306 ± 5.037 | 72.685 ± 4.923 | 72.645 ± 4.224 | 72.403 ± 3.853 | 70.500 ± 6.162 |
| 1.2 | 71.766 ± 5.117 | 72.008 ± 5.478 | 72.008 ± 5.353 | 72.718 ± 5.119 | 73.048 ± 4.601 | 72.976 ± 4.423 | 69.919 ± 6.735 |
| 2.0 | 71.589 ± 5.629 | 71.581 ± 6.115 | 71.565 ± 5.413 | 72.331 ± 5.516 | 72.887 ± 5.019 | 73.113 ± 5.005 | 68.355 ± 7.291 |
| 3.0 | 71.387 ± 5.876 | 71.927 ± 6.038 | 71.266 ± 5.837 | 72.194 ± 5.717 | 72.935 ± 5.458 | 73.065 ± 5.224 | 68.105 ± 7.742 |
| 4.0 | 71.419 ± 6.142 | 71.460 ± 5.895 | 71.177 ± 5.918 | 71.403 ± 5.795 | 72.540 ± 5.775 | 72.815 ± 5.564 | 67.847 ± 7.604 |
| 5.0 | 71.234 ± 6.367 | 71.734 ± 6.113 | 71.161 ± 6.173 | 71.387 ± 5.916 | 72.234 ± 5.867 | 72.452 ± 5.595 | 67.419 ± 8.158 |

Table 14: Results for the Ecoli dataset.

| BR | RF(nt_500) | RF(qs_ent) | RF(mn_4) | RF(mn_8) | RF(ml_4) | RF(ml_5) | RF(nf_all) |
|---|---|---|---|---|---|---|---|
| 0.2 | 84.060 ± 1.965 | 83.714 ± 2.027 | 83.295 ± 2.017 | 79.817 ± 2.070 | 76.765 ± 1.462 | 76.155 ± 1.311 | 84.161 ± 1.945 |
| 0.4 | 85.527 ± 1.799 | 84.943 ± 1.886 | 85.220 ± 1.942 | 84.589 ± 1.864 | 81.098 ± 1.966 | 78.716 ± 1.762 | 84.644 ± 1.994 |
| 0.6 | 85.835 ± 1.858 | 85.263 ± 1.928 | 85.680 ± 1.843 | 85.378 ± 1.922 | 83.496 ± 1.856 | 82.314 ± 1.962 | 84.339 ± 2.127 |
| 0.8 | 85.671 ± 1.872 | 85.164 ± 1.926 | 85.580 ± 2.000 | 85.452 ± 1.914 | 84.201 ± 1.834 | 83.573 ± 1.996 | 83.664 ± 2.242 |
| 1.0 | 85.565 ± 1.905 | 85.082 ± 1.949 | 85.589 ± 1.975 | 85.509 ± 1.924 | 84.560 ± 1.868 | 84.170 ± 1.912 | 83.049 ± 2.438 |
| 1.2 | 85.321 ± 1.981 | 84.933 ± 1.996 | 85.592 ± 1.928 | 85.554 ± 1.939 | 85.046 ± 1.895 | 84.647 ± 1.905 | 82.658 ± 2.535 |
| 2.0 | 84.859 ± 2.041 | 84.394 ± 2.073 | 85.006 ± 2.060 | 85.351 ± 2.069 | 85.106 ± 1.968 | 84.902 ± 1.945 | 81.269 ± 2.666 |
| 3.0 | 84.604 ± 2.054 | 84.104 ± 2.132 | 84.621 ± 2.049 | 84.946 ± 2.020 | 85.171 ± 2.015 | 85.070 ± 1.976 | 80.528 ± 2.765 |
| 4.0 | 84.490 ± 2.110 | 83.978 ± 2.036 | 84.439 ± 2.111 | 84.690 ± 2.096 | 85.019 ± 2.050 | 85.054 ± 1.999 | 80.137 ± 2.788 |
| 5.0 | 84.369 ± 2.111 | 83.912 ± 2.060 | 84.348 ± 2.103 | 84.568 ± 2.130 | 84.876 ± 2.011 | 84.990 ± 2.051 | 79.839 ± 2.765 |

Table 15: Results for the German Credit Data dataset.

| BR | RF(nt_500) | RF(qs_ent) | RF(mn_4) | RF(mn_8) | RF(ml_4) | RF(ml_5) | RF(nf_all) |
|---|---|---|---|---|---|---|---|
| 0.2 | 73.932 ± 1.059 | 73.797 ± 1.169 | 73.614 ± 1.169 | 73.195 ± 1.066 | 71.769 ± 0.839 | 71.227 ± 0.738 | 74.612 ± 1.322 |
| 0.4 | 75.023 ± 1.137 | 74.778 ± 1.190 | 74.700 ± 1.159 | 74.337 ± 1.176 | 73.230 ± 1.031 | 72.766 ± 0.938 | 75.046 ± 1.416 |
| 0.6 | 75.312 ± 1.110 | 75.072 ± 1.172 | 74.996 ± 1.262 | 74.816 ± 1.201 | 73.879 ± 1.090 | 73.483 ± 1.084 | 74.919 ± 1.423 |
| 0.8 | 75.410 ± 1.149 | 75.157 ± 1.199 | 75.127 ± 1.222 | 74.954 ± 1.223 | 74.276 ± 1.137 | 73.908 ± 1.113 | 74.567 ± 1.524 |
| 1.0 | 75.397 ± 1.150 | 75.195 ± 1.232 | 75.170 ± 1.197 | 75.174 ± 1.190 | 74.510 ± 1.164 | 74.178 ± 1.175 | 74.344 ± 1.615 |
| 1.2 | 75.466 ± 1.196 | 75.296 ± 1.272 | 75.262 ± 1.240 | 75.153 ± 1.242 | 74.798 ± 1.168 | 74.636 ± 1.164 | 74.127 ± 1.597 |
| 2.0 | 75.352 ± 1.196 | 75.113 ± 1.263 | 75.164 ± 1.280 | 75.154 ± 1.209 | 75.114 ± 1.217 | 75.035 ± 1.226 | 73.342 ± 1.655 |
| 3.0 | 75.375 ± 1.211 | 75.078 ± 1.209 | 75.082 ± 1.299 | 75.164 ± 1.303 | 75.324 ± 1.197 | 75.207 ± 1.191 | 72.835 ± 1.749 |
| 4.0 | 75.349 ± 1.226 | 75.073 ± 1.302 | 75.007 ± 1.288 | 75.124 ± 1.263 | 75.252 ± 1.256 | 75.249 ± 1.225 | 72.440 ± 1.781 |
| 5.0 | 75.322 ± 1.218 | 75.106 ± 1.280 | 75.110 ± 1.244 | 75.178 ± 1.279 | 75.216 ± 1.278 | 75.245 ± 1.348 | 72.175 ± 1.742 |

Table 16: Results for the Glass Identification dataset.

| BR | RF(nt_500) | RF(qs_ent) | RF(mn_4) | RF(mn_8) | RF(ml_4) | RF(ml_5) | RF(nf_all) |
|---|---|---|---|---|---|---|---|
| 0.2 | 67.159 ± 3.839 | 67.054 ± 3.664 | 65.267 ± 3.704 | 63.493 ± 3.843 | 61.211 ± 3.696 | 60.005 ± 3.950 | 65.789 ± 3.919 |
| 0.4 | 72.204 ± 3.639 | 72.366 ± 3.629 | 70.746 ± 3.679 | 67.456 ± 3.605 | 65.249 ± 3.781 | 64.087 ± 3.939 | 71.043 ± 3.915 |
| 0.6 | 74.004 ± 3.690 | 74.222 ± 3.719 | 72.821 ± 3.492 | 70.084 ± 3.738 | 67.529 ± 3.871 | 66.000 ± 3.737 | 72.572 ± 3.927 |
| 0.8 | 74.934 ± 3.675 | 74.812 ± 3.579 | 73.941 ± 3.784 | 71.793 ± 3.879 | 69.305 ± 3.829 | 67.435 ± 3.690 | 72.891 ± 3.877 |
| 1.0 | 75.250 ± 3.652 | 75.154 ± 3.608 | 74.346 ± 3.865 | 72.728 ± 3.859 | 70.676 ± 3.851 | 68.794 ± 3.761 | 72.829 ± 3.958 |
| 1.2 | 75.413 ± 3.695 | 75.424 ± 3.659 | 74.727 ± 3.768 | 73.748 ± 3.932 | 72.845 ± 3.870 | 71.479 ± 3.804 | 72.798 ± 3.983 |
| 2.0 | 75.577 ± 3.698 | 75.596 ± 3.778 | 75.176 ± 3.813 | 74.460 ± 3.889 | 74.366 ± 3.795 | 73.659 ± 3.823 | 71.466 ± 4.357 |
| 3.0 | 75.429 ± 3.751 | 75.383 ± 3.969 | 74.927 ± 3.667 | 74.819 ± 3.593 | 74.840 ± 3.687 | 74.624 ± 3.789 | 70.340 ± 4.430 |
| 4.0 | 75.328 ± 3.788 | 75.457 ± 3.842 | 74.848 ± 3.641 | 74.708 ± 3.744 | 74.868 ± 3.641 | 74.725 ± 3.698 | 69.839 ± 4.622 |
| 5.0 | 75.295 ± 3.721 | 75.363 ± 3.920 | 74.923 ± 3.736 | 74.838 ± 3.877 | 75.037 ± 3.738 | 74.931 ± 3.725 | 69.432 ± 4.705 |

Table 17: Results for the Heart dataset.

| BR | RF(nt_500) | RF(qs_ent) | RF(mn_4) | RF(mn_8) | RF(ml_4) | RF(ml_5) | RF(nf_all) |
|---|---|---|---|---|---|---|---|
| 0.2 | 83.319 ± 2.532 | 82.794 ± 2.558 | 82.969 ± 2.641 | 82.924 ± 2.685 | 83.324 ± 2.569 | 83.217 ± 2.502 | 81.669 ± 2.773 |
| 0.4 | 82.693 ± 2.339 | 82.406 ± 2.480 | 82.696 ± 2.461 | 82.806 ± 2.542 | 83.294 ± 2.553 | 83.224 ± 2.662 | 81.289 ± 2.810 |
| 0.6 | 82.272 ± 2.432 | 82.019 ± 2.495 | 82.243 ± 2.514 | 82.472 ± 2.505 | 83.137 ± 2.650 | 83.119 ± 2.584 | 80.676 ± 2.961 |
| 0.8 | 81.907 ± 2.465 | 81.800 ± 2.614 | 82.061 ± 2.490 | 82.352 ± 2.558 | 82.981 ± 2.534 | 83.148 ± 2.617 | 80.409 ± 2.981 |
| 1.0 | 81.689 ± 2.472 | 81.639 ± 2.565 | 81.846 ± 2.545 | 82.272 ± 2.470 | 82.933 ± 2.555 | 82.993 ± 2.525 | 79.961 ± 3.096 |
| 1.2 | 81.506 ± 2.501 | 81.372 ± 2.551 | 81.578 ± 2.623 | 81.917 ± 2.569 | 82.594 ± 2.550 | 82.830 ± 2.556 | 79.502 ± 2.959 |
| 2.0 | 81.046 ± 2.466 | 80.894 ± 2.481 | 80.946 ± 2.510 | 81.361 ± 2.428 | 82.093 ± 2.390 | 82.435 ± 2.453 | 78.394 ± 3.078 |
| 3.0 | 80.854 ± 2.523 | 80.707 ± 2.517 | 80.744 ± 2.465 | 81.083 ± 2.619 | 81.750 ± 2.612 | 81.989 ± 2.632 | 77.691 ± 3.164 |
| 4.0 | 80.813 ± 2.517 | 80.639 ± 2.563 | 80.680 ± 2.578 | 80.926 ± 2.610 | 81.441 ± 2.609 | 81.539 ± 2.475 | 77.250 ± 3.278 |
| 5.0 | 80.774 ± 2.541 | 80.691 ± 2.510 | 80.583 ± 2.552 | 80.835 ± 2.582 | 81.180 ± 2.652 | 81.322 ± 2.607 | 77.019 ± 3.233 |

Table 18: Results for the Hepatitis dataset.

| BR | RF(nt_500) | RF(qs_ent) | RF(mn_4) | RF(mn_8) | RF(ml_4) | RF(ml_5) | RF(nf_all) |
|---|---|---|---|---|---|---|---|
| 0.2 | 83.399 ± 2.143 | 82.936 ± 2.195 | 82.858 ± 2.107 | 82.081 ± 1.879 | 79.730 ± 0.000 | 79.730 ± 0.000 | 84.115 ± 2.303 |
| 0.4 | 84.726 ± 2.600 | 84.287 ± 2.784 | 84.149 ± 2.513 | 83.760 ± 2.454 | 80.831 ± 1.246 | 79.943 ± 0.501 | 84.044 ± 3.243 |
| 0.6 | 84.561 ± 2.587 | 84.446 ± 2.868 | 84.220 ± 2.789 | 84.189 ± 2.575 | 82.530 ± 1.869 | 81.128 ± 1.384 | 83.645 ± 3.406 |
| 0.8 | 84.355 ± 2.823 | 84.243 ± 2.997 | 84.270 ± 2.976 | 84.226 ± 2.794 | 83.439 ± 2.194 | 82.537 ± 1.866 | 83.233 ± 3.526 |
| 1.0 | 84.318 ± 2.889 | 84.216 ± 3.067 | 83.993 ± 3.007 | 84.176 ± 2.886 | 83.838 ± 2.281 | 83.236 ± 2.111 | 82.720 ± 3.387 |
| 1.2 | 84.341 ± 2.899 | 84.274 ± 2.999 | 83.993 ± 3.004 | 84.108 ± 2.986 | 84.206 ± 2.530 | 83.905 ± 2.335 | 82.274 ± 3.660 |
| 2.0 | 84.111 ± 3.048 | 84.095 ± 3.154 | 84.020 ± 3.136 | 83.970 ± 3.050 | 84.166 ± 2.772 | 84.169 ± 2.627 | 81.081 ± 3.963 |
| 3.0 | 84.020 ± 3.081 | 84.061 ± 3.223 | 83.760 ± 3.168 | 83.905 ± 3.138 | 84.020 ± 2.928 | 84.068 ± 2.862 | 80.436 ± 4.001 |
| 4.0 | 83.976 ± 3.042 | 83.895 ± 3.169 | 83.723 ± 3.068 | 83.851 ± 3.163 | 84.027 ± 2.988 | 83.983 ± 2.959 | 80.068 ± 4.158 |
| 5.0 | 83.983 ± 3.152 | 83.922 ± 3.131 | 83.672 ± 3.239 | 83.696 ± 3.054 | 83.814 ± 3.100 | 83.929 ± 2.956 | 79.959 ± 4.031 |

Table 19: Results for the Horse Colic dataset.

| BR | RF(nt_500) | RF(qs_ent) | RF(mn_4) | RF(mn_8) | RF(ml_4) | RF(ml_5) | RF(nf_all) |
|---|---|---|---|---|---|---|---|
| 0.2 | 85.883 ± 1.902 | 85.681 ± 1.959 | 85.645 ± 1.848 | 85.685 ± 1.904 | 84.041 ± 2.054 | 82.474 ± 2.331 | 84.861 ± 2.541 |
| 0.4 | 86.111 ± 1.882 | 85.933 ± 1.905 | 86.056 ± 1.812 | 86.090 ± 1.837 | 85.670 ± 1.840 | 85.458 ± 1.822 | 86.052 ± 2.245 |
| 0.6 | 86.285 ± 1.872 | 86.240 ± 1.858 | 86.193 ± 1.855 | 86.230 ± 1.843 | 85.780 ± 1.860 | 85.698 ± 1.878 | 86.269 ± 2.141 |
| 0.8 | 86.484 ± 1.879 | 86.279 ± 1.903 | 86.315 ± 1.932 | 86.216 ± 1.881 | 85.815 ± 1.907 | 85.774 ± 1.798 | 86.095 ± 2.068 |
| 1.0 | 86.516 ± 1.918 | 86.311 ± 1.852 | 86.303 ± 1.808 | 86.197 ± 1.805 | 85.818 ± 1.927 | 85.766 ± 1.826 | 85.769 ± 2.199 |
| 1.2 | 86.515 ± 1.888 | 86.365 ± 1.855 | 86.371 ± 1.910 | 86.295 ± 1.827 | 85.856 ± 1.839 | 85.780 ± 1.875 | 85.387 ± 2.231 |
| 2.0 | 86.438 ± 1.871 | 86.226 ± 1.920 | 86.360 ± 1.873 | 86.383 ± 1.901 | 85.887 ± 1.998 | 85.781 ± 1.962 | 84.645 ± 2.301 |
| 3.0 | 86.480 ± 1.872 | 86.202 ± 1.919 | 86.306 ± 1.886 | 86.421 ± 1.902 | 86.144 ± 1.953 | 85.950 ± 1.870 | 84.007 ± 2.395 |
| 4.0 | 86.395 ± 1.844 | 86.148 ± 1.928 | 86.156 ± 1.915 | 86.205 ± 1.865 | 86.167 ± 1.926 | 86.163 ± 1.899 | 83.776 ± 2.420 |
| 5.0 | 86.414 ± 1.846 | 86.177 ± 1.797 | 86.261 ± 1.926 | 86.276 ± 1.861 | 86.216 ± 1.936 | 86.166 ± 1.941 | 83.569 ± 2.373 |

Table 20: Results for the Image Segmentation (Stat.) dataset.

| BR | RF(nt_500) | RF(qs_ent) | RF(mn_4) | RF(mn_8) | RF(ml_4) | RF(ml_5) | RF(nf_all) |
|---|---|---|---|---|---|---|---|
| 0.2 | 95.150 ± 0.754 | 95.081 ± 0.718 | 94.914 ± 0.783 | 94.360 ± 0.816 | 93.540 ± 0.785 | 93.179 ± 0.774 | 95.126 ± 0.739 |
| 0.4 | 96.213 ± 0.583 | 96.143 ± 0.605 | 96.049 ± 0.615 | 95.634 ± 0.679 | 94.868 ± 0.759 | 94.518 ± 0.798 | 95.859 ± 0.662 |
| 0.6 | 96.511 ± 0.539 | 96.502 ± 0.577 | 96.418 ± 0.555 | 96.130 ± 0.581 | 95.465 ± 0.681 | 95.151 ± 0.722 | 96.070 ± 0.612 |
| 0.8 | 96.712 ± 0.528 | 96.718 ± 0.559 | 96.608 ± 0.548 | 96.365 ± 0.547 | 95.802 ± 0.645 | 95.519 ± 0.665 | 96.150 ± 0.621 |
| 1.0 | 96.825 ± 0.509 | 96.843 ± 0.563 | 96.728 ± 0.544 | 96.498 ± 0.564 | 95.996 ± 0.625 | 95.720 ± 0.663 | 96.184 ± 0.631 |
| 1.2 | 96.898 ± 0.513 | 96.924 ± 0.522 | 96.814 ± 0.528 | 96.680 ± 0.549 | 96.328 ± 0.589 | 96.104 ± 0.618 | 96.190 ± 0.625 |
| 2.0 | 97.018 ± 0.512 | 97.071 ± 0.539 | 96.967 ± 0.524 | 96.876 ± 0.529 | 96.604 ± 0.552 | 96.479 ± 0.548 | 96.094 ± 0.662 |
| 3.0 | 97.064 ± 0.525 | 97.107 ± 0.522 | 96.989 ± 0.532 | 96.993 ± 0.524 | 96.805 ± 0.548 | 96.702 ± 0.553 | 95.970 ± 0.669 |
| 4.0 | 97.081 ± 0.521 | 97.130 ± 0.544 | 97.023 ± 0.523 | 97.017 ± 0.537 | 96.906 ± 0.539 | 96.824 ± 0.525 | 95.842 ± 0.690 |
| 5.0 | 97.080 ± 0.530 | 97.133 ± 0.538 | 97.018 ± 0.538 | 97.013 ± 0.544 | 96.939 ± 0.540 | 96.880 ± 0.554 | 95.773 ± 0.692 |

Table 21: Results for the Ionosphere dataset.

| BR | RF(nt_500) | RF(qs_ent) | RF(mn_4) | RF(mn_8) | RF(ml_4) | RF(ml_5) | RF(nf_all) |
|---|---|---|---|---|---|---|---|
| 0.2 | 92.629 ± 1.511 | 91.544 ± 1.839 | 91.913 ± 1.651 | 91.934 ± 1.701 | 88.669 ± 2.406 | 86.764 ± 2.230 | 91.700 ± 1.979 |
| 0.4 | 93.076 ± 1.448 | 92.580 ± 1.591 | 92.643 ± 1.533 | 92.511 ± 1.575 | 91.760 ± 1.758 | 91.183 ± 1.996 | 92.256 ± 2.013 |
| 0.6 | 93.194 ± 1.495 | 92.753 ± 1.513 | 92.699 ± 1.550 | 92.666 ± 1.538 | 92.253 ± 1.611 | 91.986 ± 1.685 | 92.190 ± 2.083 |
| 0.8 | 93.209 ± 1.475 | 92.937 ± 1.544 | 92.829 ± 1.547 | 92.707 ± 1.561 | 92.443 ± 1.574 | 92.310 ± 1.635 | 91.961 ± 2.139 |
| 1.0 | 93.231 ± 1.524 | 93.024 ± 1.499 | 92.881 ± 1.493 | 92.764 ± 1.564 | 92.591 ± 1.587 | 92.400 ± 1.593 | 91.730 ± 2.169 |
| 1.2 | 93.254 ± 1.526 | 93.020 ± 1.588 | 92.903 ± 1.637 | 92.814 ± 1.613 | 92.776 ± 1.586 | 92.630 ± 1.593 | 91.571 ± 2.190 |
| 2.0 | 93.233 ± 1.595 | 93.070 ± 1.556 | 92.957 ± 1.640 | 92.851 ± 1.627 | 92.843 ± 1.572 | 92.770 ± 1.592 | 90.896 ± 2.281 |
| 3.0 | 93.229 ± 1.600 | 93.099 ± 1.608 | 93.003 ± 1.614 | 92.906 ± 1.615 | 92.897 ± 1.594 | 92.836 ± 1.572 | 90.474 ± 2.443 |
| 4.0 | 93.240 ± 1.598 | 93.103 ± 1.523 | 93.010 ± 1.632 | 92.956 ± 1.709 | 92.844 ± 1.627 | 92.881 ± 1.597 | 90.151 ± 2.442 |
| 5.0 | 93.237 ± 1.571 | 93.127 ± 1.577 | 93.039 ± 1.615 | 92.964 ± 1.625 | 92.911 ± 1.604 | 92.927 ± 1.559 | 89.920 ± 2.498 |

Table 22: Results for the Iris dataset.

| BR | RF(nt_500) | RF(qs_ent) | RF(mn_4) | RF(mn_8) | RF(ml_4) | RF(ml_5) | RF(nf_all) |
|---|---|---|---|---|---|---|---|
| 0.2 | 95.178 ± 1.824 | 95.098 ± 1.833 | 95.171 ± 1.876 | 95.199 ± 1.899 | 95.095 ± 1.951 | 94.880 ± 2.076 | 95.218 ± 1.840 |
| 0.4 | 95.115 ± 1.831 | 95.058 ± 1.804 | 95.232 ± 1.900 | 95.185 ± 1.878 | 95.155 ± 1.835 | 95.212 ± 1.860 | 95.084 ± 1.856 |
| 0.6 | 95.010 ± 1.870 | 94.947 ± 1.863 | 95.145 ± 1.941 | 95.185 ± 1.983 | 95.081 ± 1.923 | 95.081 ± 1.949 | 95.017 ± 1.901 |
| 0.8 | 94.967 ± 1.892 | 94.859 ± 1.883 | 95.091 ± 1.982 | 95.131 ± 2.049 | 94.963 ± 1.986 | 94.947 ± 1.980 | 94.940 ± 1.926 |
| 1.0 | 94.950 ± 1.904 | 94.863 ± 1.915 | 95.064 ± 1.972 | 95.101 ± 2.033 | 94.920 ± 2.030 | 94.859 ± 1.989 | 94.886 ± 1.947 |
| 1.2 | 94.917 ± 1.890 | 94.860 ± 1.931 | 95.020 ± 1.977 | 95.138 ± 2.050 | 94.919 ± 2.036 | 94.896 ± 2.015 | 94.883 ± 1.989 |
| 2.0 | 94.863 ± 1.919 | 94.826 ± 1.915 | 94.933 ± 1.942 | 95.155 ± 2.053 | 94.943 ± 2.067 | 94.876 ± 2.039 | 94.624 ± 2.239 |
| 3.0 | 94.883 ± 1.893 | 94.819 ± 1.865 | 94.842 ± 1.929 | 95.054 ± 1.989 | 94.947 ± 2.000 | 95.001 ± 2.102 | 94.517 ± 2.294 |
| 4.0 | 94.839 ± 1.898 | 94.816 ± 1.928 | 94.812 ± 1.916 | 94.964 ± 1.959 | 94.897 ± 1.976 | 94.954 ± 2.050 | 94.517 ± 2.223 |
| 5.0 | 94.849 ± 1.914 | 94.806 ± 1.895 | 94.792 ± 1.899 | 94.890 ± 1.955 | 94.833 ± 1.946 | 94.897 ± 1.981 | 94.500 ± 2.237 |

Table 23: Results for the Labor Relations dataset.

| BR | RF(nt_500) | RF(qs_ent) | RF(mn_4) | RF(mn_8) | RF(ml_4) | RF(ml_5) | RF(nf_all) |
|---|---|---|---|---|---|---|---|
| 0.2 | 82.500 ± 5.787 | 83.768 ± 6.384 | 80.735 ± 6.456 | 64.901 ± 0.616 | 64.901 ± 0.616 | 64.901 ± 0.616 | 84.817 ± 6.153 |
| 0.4 | 90.273 ± 5.306 | 90.113 ± 5.135 | 89.246 ± 5.120 | 84.780 ± 5.871 | 66.436 ± 2.657 | 64.901 ± 0.616 | 88.058 ± 5.846 |
| 0.6 | 92.629 ± 4.660 | 92.268 ± 4.704 | 91.790 ± 4.777 | 90.437 ± 4.619 | 78.279 ± 5.683 | 70.293 ± 4.840 | 89.736 ± 5.886 |
| 0.8 | 93.227 ± 4.407 | 92.929 ± 4.537 | 92.682 ± 4.575 | 91.825 ± 4.584 | 83.577 ± 5.449 | 77.870 ± 5.680 | 90.354 ± 5.748 |
| 1.0 | 93.381 ± 4.478 | 92.865 ± 4.579 | 92.837 ± 4.656 | 92.354 ± 4.416 | 85.311 ± 5.239 | 80.514 ± 5.594 | 90.369 ± 5.799 |
| 1.2 | 93.608 ± 4.329 | 93.374 ± 4.445 | 93.339 ± 4.390 | 92.699 ± 4.447 | 89.673 ± 4.926 | 87.363 ± 5.051 | 89.845 ± 6.026 |
| 2.0 | 93.522 ± 4.419 | 93.408 ± 4.362 | 93.346 ± 4.447 | 93.222 ± 4.362 | 91.798 ± 4.772 | 90.671 ± 4.840 | 88.915 ± 6.119 |
| 3.0 | 93.445 ± 4.414 | 93.332 ± 4.374 | 93.191 ± 4.356 | 93.127 ± 4.388 | 92.531 ± 4.538 | 92.169 ± 4.463 | 88.321 ± 6.562 |
| 4.0 | 93.496 ± 4.366 | 93.365 ± 4.392 | 93.375 ± 4.313 | 93.215 ± 4.309 | 93.065 ± 4.282 | 92.723 ± 4.303 | 87.866 ± 6.702 |
| 5.0 | 93.462 ± 4.390 | 93.304 ± 4.448 | 93.127 ± 4.415 | 93.111 ± 4.468 | 93.077 ± 4.445 | 92.750 ± 4.578 | 87.597 ± 6.552 |

Table 24: Results for the Liver Disorders dataset.

| BR | RF(nt_500) | RF(qs_ent) | RF(mn_4) | RF(mn_8) | RF(ml_4) | RF(ml_5) | RF(nf_all) |
|---|---|---|---|---|---|---|---|
| 0.2 | 59.507 ± 2.770 | 59.089 ± 2.991 | 59.164 ± 2.827 | 59.485 ± 3.014 | 59.714 ± 2.951 | 59.711 ± 2.870 | 59.213 ± 2.791 |
| 0.4 | 58.570 ± 2.799 | 58.240 ± 2.868 | 58.406 ± 2.697 | 58.757 ± 2.760 | 59.109 ± 2.816 | 59.209 ± 2.880 | 58.219 ± 2.778 |
| 0.6 | 57.910 ± 2.817 | 57.616 ± 2.845 | 57.923 ± 3.027 | 58.482 ± 2.786 | 58.921 ± 2.846 | 59.206 ± 2.851 | 57.605 ± 2.928 |
| 0.8 | 57.391 ± 2.867 | 57.352 ± 2.883 | 57.752 ± 2.896 | 58.259 ± 2.828 | 58.825 ± 2.831 | 59.186 ± 2.841 | 56.903 ± 2.998 |
| 1.0 | 57.016 ± 2.746 | 57.061 ± 3.103 | 57.387 ± 2.934 | 57.916 ± 2.934 | 58.635 ± 2.860 | 58.991 ± 2.877 | 56.614 ± 2.884 |
| 1.2 | 56.752 ± 2.880 | 56.638 ± 2.901 | 56.961 ± 2.938 | 57.399 ± 2.883 | 58.183 ± 2.858 | 58.446 ± 2.739 | 56.409 ± 3.139 |
| 2.0 | 56.160 ± 2.936 | 55.975 ± 3.188 | 56.290 ± 3.053 | 56.956 ± 2.935 | 57.720 ± 2.865 | 57.935 ± 2.867 | 55.569 ± 3.149 |
| 3.0 | 55.751 ± 3.008 | 55.612 ± 3.181 | 55.826 ± 3.100 | 56.144 ± 3.027 | 56.830 ± 2.886 | 57.326 ± 3.121 | 55.075 ± 3.227 |
| 4.0 | 55.755 ± 3.067 | 55.578 ± 3.134 | 55.721 ± 3.039 | 55.867 ± 3.113 | 56.548 ± 3.034 | 56.877 ± 3.091 | 54.922 ± 3.159 |
| 5.0 | 55.645 ± 3.090 | 55.452 ± 3.043 | 55.755 ± 3.006 | 55.776 ± 3.166 | 56.301 ± 3.055 | 56.460 ± 2.977 | 54.812 ± 3.189 |

Table 25: Results for the Optical Recognition (Digits) dataset.

| BR | RF(nt_500) | RF(qs_ent) | RF(mn_4) | RF(mn_8) | RF(ml_4) | RF(ml_5) | RF(nf_all) |
|---|---|---|---|---|---|---|---|
| 0.2 | 95.549 ± 0.677 | 95.168 ± 0.742 | 94.761 ± 0.739 | 94.161 ± 0.814 | 93.207 ± 0.878 | 92.702 ± 0.884 | 93.174 ± 0.985 |
| 0.4 | 96.386 ± 0.593 | 96.156 ± 0.642 | 95.861 ± 0.679 | 95.341 ± 0.759 | 94.507 ± 0.799 | 94.029 ± 0.788 | 93.866 ± 0.975 |
| 0.6 | 96.744 ± 0.572 | 96.553 ± 0.617 | 96.310 ± 0.623 | 95.873 ± 0.664 | 95.094 ± 0.740 | 94.691 ± 0.747 | 94.074 ± 0.977 |
| 0.8 | 96.964 ± 0.562 | 96.811 ± 0.544 | 96.581 ± 0.571 | 96.188 ± 0.627 | 95.488 ± 0.713 | 95.104 ± 0.759 | 94.094 ± 1.018 |
| 1.0 | 97.108 ± 0.548 | 96.933 ± 0.539 | 96.741 ± 0.580 | 96.378 ± 0.633 | 95.717 ± 0.713 | 95.387 ± 0.711 | 94.014 ± 1.045 |
| 1.2 | 97.176 ± 0.535 | 96.986 ± 0.512 | 96.883 ± 0.572 | 96.624 ± 0.598 | 96.108 ± 0.634 | 95.849 ± 0.684 | 93.943 ± 1.061 |
| 2.0 | 97.340 ± 0.524 | 97.194 ± 0.499 | 97.085 ± 0.564 | 96.898 ± 0.584 | 96.536 ± 0.598 | 96.267 ± 0.642 | 93.335 ± 1.209 |
| 3.0 | 97.396 ± 0.505 | 97.246 ± 0.531 | 97.208 ± 0.527 | 97.096 ± 0.555 | 96.794 ± 0.596 | 96.654 ± 0.601 | 92.561 ± 1.423 |
| 4.0 | 97.413 ± 0.519 | 97.255 ± 0.514 | 97.218 ± 0.540 | 97.153 ± 0.556 | 96.977 ± 0.591 | 96.858 ± 0.599 | 91.948 ± 1.544 |
| 5.0 | 97.401 ± 0.514 | 97.263 ± 0.511 | 97.233 ± 0.539 | 97.192 ± 0.530 | 97.050 ± 0.541 | 96.944 ± 0.581 | 91.474 ± 1.645 |

Table 26: Results for the Parkinsons dataset.

| BR | RF(nt_500) | RF(qs_ent) | RF(mn_4) | RF(mn_8) | RF(ml_4) | RF(ml_5) | RF(nf_all) |
|---|---|---|---|---|---|---|---|
| 0.2 | 84.367 ± 3.323 | 84.487 ± 3.351 | 83.902 ± 3.371 | 82.746 ± 3.191 | 81.979 ± 3.268 | 81.174 ± 3.147 | 84.482 ± 3.362 |
| 0.4 | 86.205 ± 3.474 | 86.203 ± 3.337 | 85.618 ± 3.554 | 84.339 ± 3.385 | 84.187 ± 3.226 | 83.464 ± 3.141 | 85.950 ± 3.506 |
| 0.6 | 87.223 ± 3.458 | 87.413 ± 3.426 | 86.705 ± 3.585 | 85.462 ± 3.544 | 84.993 ± 3.256 | 84.446 ± 3.294 | 86.554 ± 3.622 |
| 0.8 | 87.949 ± 3.369 | 88.085 ± 3.256 | 87.408 ± 3.375 | 86.336 ± 3.557 | 85.757 ± 3.364 | 85.097 ± 3.383 | 86.828 ± 3.765 |
| 1.0 | 88.297 ± 3.317 | 88.354 ± 3.123 | 87.874 ± 3.369 | 86.674 ± 3.516 | 86.275 ± 3.346 | 85.687 ± 3.400 | 87.151 ± 3.840 |
| 1.2 | 88.536 ± 3.299 | 88.736 ± 3.191 | 88.426 ± 3.260 | 87.629 ± 3.330 | 87.069 ± 3.297 | 86.528 ± 3.318 | 87.195 ± 3.914 |
| 2.0 | 89.093 ± 3.270 | 89.150 ± 3.270 | 88.854 ± 3.136 | 88.462 ± 3.320 | 87.782 ± 3.336 | 87.418 ± 3.404 | 87.335 ± 4.205 |
| 3.0 | 89.213 ± 3.262 | 89.244 ± 3.162 | 89.139 ± 3.216 | 88.877 ± 3.245 | 88.482 ± 3.285 | 88.215 ± 3.268 | 87.167 ± 4.156 |
| 4.0 | 89.306 ± 3.254 | 89.206 ± 3.208 | 89.154 ± 3.123 | 89.031 ± 3.269 | 88.628 ± 3.298 | 88.508 ± 3.275 | 87.000 ± 4.281 |
| 5.0 | 89.306 ± 3.234 | 89.160 ± 3.296 | 89.213 ± 3.070 | 89.133 ± 3.130 | 88.810 ± 3.169 | 88.710 ± 3.194 | 86.941 ± 4.166 |

Table 27: Results for the Pima Indians Diabetes dataset.

| BR | RF(nt_500) | RF(qs_ent) | RF(mn_4) | RF(mn_8) | RF(ml_4) | RF(ml_5) | RF(nf_all) |
|---|---|---|---|---|---|---|---|
| 0.2 | 76.344 ± 1.557 | 75.767 ± 1.632 | 76.092 ± 1.559 | 76.144 ± 1.580 | 75.973 ± 1.505 | 75.805 ± 1.522 | 76.333 ± 1.659 |
| 0.4 | 76.285 ± 1.662 | 76.020 ± 1.555 | 76.152 ± 1.575 | 76.086 ± 1.588 | 76.061 ± 1.561 | 76.040 ± 1.543 | 75.995 ± 1.631 |
| 0.6 | 76.161 ± 1.619 | 76.022 ± 1.647 | 76.069 ± 1.552 | 76.099 ± 1.592 | 76.101 ± 1.578 | 75.965 ± 1.555 | 75.912 ± 1.636 |
| 0.8 | 76.071 ± 1.633 | 75.937 ± 1.609 | 75.998 ± 1.596 | 76.100 ± 1.711 | 76.058 ± 1.620 | 76.048 ± 1.564 | 75.566 ± 1.743 |
| 1.0 | 75.950 ± 1.658 | 75.752 ± 1.627 | 75.921 ± 1.681 | 75.984 ± 1.690 | 76.061 ± 1.636 | 76.148 ± 1.607 | 75.316 ± 1.707 |
| 1.2 | 75.883 ± 1.626 | 75.745 ± 1.648 | 75.764 ± 1.648 | 75.999 ± 1.667 | 76.096 ± 1.649 | 76.075 ± 1.616 | 75.107 ± 1.727 |
| 2.0 | 75.736 ± 1.648 | 75.579 ± 1.617 | 75.549 ± 1.693 | 75.736 ± 1.733 | 75.937 ± 1.637 | 75.987 ± 1.660 | 74.475 ± 1.832 |
| 3.0 | 75.500 ± 1.642 | 75.386 ± 1.675 | 75.349 ± 1.728 | 75.578 ± 1.731 | 75.821 ± 1.641 | 75.811 ± 1.680 | 74.051 ± 1.878 |
| 4.0 | 75.517 ± 1.684 | 75.370 ± 1.714 | 75.281 ± 1.667 | 75.446 ± 1.720 | 75.598 ± 1.613 | 75.739 ± 1.636 | 73.600 ± 1.846 |
| 5.0 | 75.449 ± 1.700 | 75.231 ± 1.705 | 75.320 ± 1.663 | 75.378 ± 1.669 | 75.575 ± 1.635 | 75.602 ± 1.705 | 73.443 ± 1.910 |

Table 28: Results for the Sonar, Mines vs. Rocks dataset.

| BR | RF(nt_500) | RF(qs_ent) | RF(mn_4) | RF(mn_8) | RF(ml_4) | RF(ml_5) | RF(nf_all) |
|---|---|---|---|---|---|---|---|
| 0.2 | 77.678 ± 3.731 | 76.805 ± 3.706 | 76.656 ± 3.867 | 76.327 ± 3.705 | 76.135 ± 3.647 | 75.899 ± 3.627 | 75.820 ± 3.805 |
| 0.4 | 78.594 ± 3.922 | 78.212 ± 3.900 | 77.959 ± 3.892 | 77.404 ± 3.868 | 77.752 ± 3.845 | 77.351 ± 3.948 | 76.356 ± 3.951 |
| 0.6 | 79.120 ± 3.966 | 79.353 ± 4.033 | 78.697 ± 4.141 | 78.159 ± 3.992 | 78.433 ± 4.140 | 78.099 ± 4.079 | 76.887 ± 4.058 |
| 0.8 | 79.572 ± 3.961 | 79.870 ± 3.937 | 79.209 ± 4.078 | 78.442 ± 4.099 | 78.623 ± 4.087 | 78.418 ± 4.111 | 76.803 ± 4.042 |
| 1.0 | 80.214 ± 3.992 | 80.240 ± 3.903 | 79.625 ± 4.093 | 78.894 ± 4.004 | 79.130 ± 4.014 | 78.808 ± 4.019 | 76.897 ± 4.227 |
| 1.2 | 80.618 ± 4.143 | 80.053 ± 3.981 | 79.750 ± 4.055 | 79.296 ± 4.040 | 79.397 ± 4.041 | 79.055 ± 3.962 | 76.894 ± 4.261 |
| 2.0 | 81.236 ± 4.060 | 81.094 ± 3.975 | 80.579 ± 4.141 | 80.276 ± 4.192 | 80.091 ± 4.045 | 79.817 ± 4.206 | 76.548 ± 4.440 |
| 3.0 | 81.486 ± 4.057 | 81.308 ± 4.024 | 80.974 ± 3.884 | 80.656 ± 3.927 | 80.514 ± 4.021 | 80.341 ± 4.103 | 75.822 ± 4.701 |
| 4.0 | 81.565 ± 4.072 | 81.627 ± 4.007 | 81.317 ± 4.061 | 81.113 ± 4.037 | 80.962 ± 3.934 | 80.668 ± 3.834 | 75.464 ± 4.878 |
| 5.0 | 81.548 ± 4.161 | 81.538 ± 3.989 | 81.012 ± 4.123 | 81.161 ± 3.924 | 80.930 ± 3.939 | 80.913 ± 4.007 | 74.950 ± 4.863 |

Table 29: Results for the Soybean (Large) dataset.

| BR | RF(nt_500) | RF(qs_ent) | RF(mn_4) | RF(mn_8) | RF(ml_4) | RF(ml_5) | RF(nf_all) |
|---|---|---|---|---|---|---|---|
| 0.2 | 90.450 ± 1.379 | 89.572 ± 1.557 | 89.425 ± 1.555 | 86.323 ± 1.798 | 74.294 ± 2.680 | 67.276 ± 2.540 | 89.222 ± 1.595 |
| 0.4 | 91.692 ± 1.346 | 91.319 ± 1.321 | 91.410 ± 1.349 | 90.479 ± 1.479 | 85.207 ± 1.979 | 82.423 ± 2.141 | 91.089 ± 1.541 |
| 0.6 | 92.186 ± 1.312 | 91.806 ± 1.382 | 92.050 ± 1.336 | 91.511 ± 1.395 | 87.737 ± 1.699 | 85.760 ± 1.888 | 91.463 ± 1.516 |
| 0.8 | 92.386 ± 1.280 | 92.013 ± 1.288 | 92.268 ± 1.291 | 91.938 ± 1.319 | 88.928 ± 1.636 | 87.261 ± 1.727 | 91.628 ± 1.467 |
| 1.0 | 92.442 ± 1.282 | 92.101 ± 1.308 | 92.459 ± 1.327 | 92.204 ± 1.316 | 89.641 ± 1.588 | 88.136 ± 1.732 | 91.664 ± 1.474 |
| 1.2 | 92.521 ± 1.296 | 92.233 ± 1.293 | 92.440 ± 1.294 | 92.479 ± 1.278 | 90.923 ± 1.434 | 90.044 ± 1.516 | 91.635 ± 1.488 |
| 2.0 | 92.606 ± 1.299 | 92.348 ± 1.254 | 92.598 ± 1.304 | 92.695 ± 1.263 | 91.704 ± 1.355 | 91.182 ± 1.450 | 91.505 ± 1.509 |
| 3.0 | 92.641 ± 1.300 | 92.412 ± 1.296 | 92.618 ± 1.292 | 92.691 ± 1.240 | 92.153 ± 1.340 | 91.985 ± 1.351 | 91.274 ± 1.564 |
| 4.0 | 92.640 ± 1.251 | 92.480 ± 1.278 | 92.561 ± 1.297 | 92.712 ± 1.276 | 92.403 ± 1.278 | 92.220 ± 1.327 | 91.133 ± 1.613 |
| 5.0 | 92.662 ± 1.246 | 92.537 ± 1.288 | 92.566 ± 1.290 | 92.681 ± 1.314 | 92.526 ± 1.291 | 92.387 ± 1.289 | 91.027 ± 1.623 |

Table 30: Results for the Tic-Tac-Toe Endgame dataset.

| BR | RF(nt_500) | RF(qs_ent) | RF(mn_4) | RF(mn_8) | RF(ml_4) | RF(ml_5) | RF(nf_all) |
|---|---|---|---|---|---|---|---|
| 0.2 | 84.737 ± 1.551 | 84.186 ± 1.683 | 82.142 ± 1.618 | 78.510 ± 1.538 | 76.485 ± 1.409 | 75.429 ± 1.290 | 88.951 ± 2.208 |
| 0.4 | 90.749 ± 1.738 | 90.185 ± 1.796 | 88.007 ± 1.783 | 83.849 ± 1.626 | 81.011 ± 1.574 | 79.460 ± 1.552 | 95.987 ± 1.283 |
| 0.6 | 93.423 ± 1.661 | 92.680 ± 1.595 | 90.843 ± 1.747 | 86.760 ± 1.740 | 83.504 ± 1.666 | 81.897 ± 1.641 | 96.907 ± 1.053 |
| 0.8 | 94.821 ± 1.502 | 93.998 ± 1.537 | 92.594 ± 1.689 | 88.705 ± 1.812 | 85.392 ± 1.809 | 83.569 ± 1.730 | 97.150 ± 0.981 |
| 1.0 | 95.590 ± 1.387 | 95.009 ± 1.396 | 93.755 ± 1.592 | 90.200 ± 1.807 | 86.947 ± 1.827 | 85.031 ± 1.774 | 97.131 ± 1.043 |
| 1.2 | 96.101 ± 1.310 | 95.585 ± 1.366 | 94.832 ± 1.546 | 92.697 ± 1.652 | 89.772 ± 1.873 | 88.176 ± 1.851 | 97.110 ± 1.027 |
| 2.0 | 96.866 ± 1.174 | 96.424 ± 1.192 | 96.055 ± 1.252 | 94.711 ± 1.475 | 92.581 ± 1.781 | 91.113 ± 1.852 | 96.676 ± 1.229 |
| 3.0 | 97.162 ± 1.094 | 96.697 ± 1.118 | 96.537 ± 1.200 | 95.793 ± 1.317 | 94.310 ± 1.567 | 93.478 ± 1.713 | 96.235 ± 1.312 |
| 4.0 | 97.184 ± 1.063 | 96.819 ± 1.113 | 96.778 ± 1.088 | 96.333 ± 1.246 | 95.201 ± 1.467 | 94.563 ± 1.573 | 95.908 ± 1.438 |
| 5.0 | 97.264 ± 1.065 | 96.858 ± 1.128 | 96.873 ± 1.104 | 96.545 ± 1.172 | 95.831 ± 1.306 | 95.226 ± 1.429 | 95.608 ± 1.482 |

Table 31: Results for the Thyroid Disease dataset.

| BR | RF(nt_500) | RF(qs_ent) | RF(mn_4) | RF(mn_8) | RF(ml_4) | RF(ml_5) | RF(nf_all) |
|---|---|---|---|---|---|---|---|
| 0.2 | 92.821 ± 2.404 | 92.992 ± 2.334 | 93.078 ± 2.393 | 92.924 ± 2.317 | 89.785 ± 2.068 | 82.280 ± 3.670 | 93.146 ± 2.386 |
| 0.4 | 95.148 ± 1.951 | 94.949 ± 1.940 | 95.049 ± 2.068 | 94.822 ± 2.099 | 93.696 ± 2.220 | 93.371 ± 2.124 | 93.927 ± 2.296 |
| 0.6 | 95.529 ± 1.952 | 95.478 ± 1.831 | 95.384 ± 2.054 | 95.166 ± 2.103 | 94.431 ± 2.199 | 94.229 ± 2.256 | 94.203 ± 2.366 |
| 0.8 | 95.552 ± 1.945 | 95.675 ± 1.848 | 95.366 ± 1.938 | 95.087 ± 2.067 | 94.631 ± 2.131 | 94.496 ± 2.179 | 94.142 ± 2.378 |
| 1.0 | 95.542 ± 1.926 | 95.766 ± 1.841 | 95.331 ± 1.981 | 95.070 ± 2.055 | 94.710 ± 2.176 | 94.531 ± 2.210 | 94.080 ± 2.362 |
| 1.2 | 95.624 ± 1.903 | 95.840 ± 1.777 | 95.482 ± 1.917 | 95.240 ± 1.968 | 95.010 ± 2.091 | 94.870 ± 2.109 | 94.078 ± 2.288 |
| 2.0 | 95.459 ± 1.935 | 95.635 ± 1.858 | 95.328 ± 1.924 | 95.112 ± 2.057 | 94.964 ± 2.034 | 94.815 ± 2.110 | 93.763 ± 2.308 |
| 3.0 | 95.321 ± 2.021 | 95.565 ± 1.853 | 95.152 ± 2.030 | 95.029 ± 2.010 | 94.882 ± 2.139 | 94.778 ± 2.179 | 93.689 ± 2.327 |
| 4.0 | 95.217 ± 2.047 | 95.523 ± 1.882 | 95.042 ± 2.059 | 94.924 ± 2.080 | 94.847 ± 2.193 | 94.768 ± 2.280 | 93.622 ± 2.378 |
| 5.0 | 95.163 ± 2.044 | 95.423 ± 1.924 | 94.977 ± 2.058 | 94.898 ± 2.041 | 94.858 ± 2.108 | 94.724 ± 2.113 | 93.601 ± 2.347 |

Table 32: Results for the Vehicle Silhouettes dataset.

| BR | RF(nt_500) | RF(qs_ent) | RF(mn_4) | RF(mn_8) | RF(ml_4) | RF(ml_5) | RF(nf_all) |
|---|---|---|---|---|---|---|---|
| 0.2 | 72.444 ± 1.686 | 72.441 ± 1.739 | 72.170 ± 1.746 | 71.441 ± 1.810 | 70.228 ± 1.823 | 69.615 ± 1.790 | 72.351 ± 1.817 |
| 0.4 | 73.467 ± 1.615 | 73.362 ± 1.587 | 73.238 ± 1.683 | 72.954 ± 1.710 | 71.817 ± 1.768 | 71.322 ± 1.774 | 73.385 ± 1.837 |
| 0.6 | 73.932 ± 1.618 | 73.790 ± 1.564 | 73.845 ± 1.625 | 73.434 ± 1.670 | 72.490 ± 1.695 | 71.973 ± 1.794 | 73.595 ± 1.680 |
| 0.8 | 74.286 ± 1.606 | 74.070 ± 1.560 | 74.118 ± 1.673 | 73.737 ± 1.597 | 72.794 ± 1.692 | 72.355 ± 1.770 | 73.597 ± 1.682 |
| 1.0 | 74.397 ± 1.591 | 74.227 ± 1.509 | 74.303 ± 1.614 | 73.998 ± 1.663 | 73.005 ± 1.694 | 72.568 ± 1.683 | 73.656 ± 1.709 |
| 1.2 | 74.442 ± 1.581 | 74.322 ± 1.568 | 74.347 ± 1.608 | 74.212 ± 1.590 | 73.451 ± 1.695 | 73.160 ± 1.646 | 73.471 ± 1.747 |
| 2.0 | 74.557 ± 1.519 | 74.487 ± 1.561 | 74.566 ± 1.615 | 74.422 ± 1.555 | 73.948 ± 1.619 | 73.616 ± 1.683 | 73.220 ± 1.765 |
| 3.0 | 74.506 ± 1.542 | 74.434 ± 1.576 | 74.531 ± 1.562 | 74.455 ± 1.597 | 74.127 ± 1.618 | 74.071 ± 1.574 | 72.681 ± 1.856 |
| 4.0 | 74.544 ± 1.480 | 74.547 ± 1.582 | 74.480 ± 1.529 | 74.522 ± 1.562 | 74.214 ± 1.536 | 74.138 ± 1.563 | 72.343 ± 1.885 |
| 5.0 | 74.583 ± 1.512 | 74.442 ± 1.576 | 74.544 ± 1.577 | 74.505 ± 1.549 | 74.326 ± 1.665 | 74.276 ± 1.495 | 72.091 ± 1.918 |

Table 33: Results for the Vowel Recognition dataset.

| BR | RF(nt_500) | RF(qs_ent) | RF(mn_4) | RF(mn_8) | RF(ml_4) | RF(ml_5) | RF(nf_all) |
|---|---|---|---|---|---|---|---|
| 0.2 | 84.796 ± 2.104 | 81.707 ± 2.151 | 80.631 ± 2.235 | 76.111 ± 2.369 | 73.027 ± 2.420 | 70.847 ± 2.581 | 77.066 ± 2.491 |
| 0.4 | 88.552 ± 1.902 | 86.939 ± 2.015 | 85.794 ± 2.090 | 82.279 ± 2.149 | 78.848 ± 2.342 | 76.617 ± 2.353 | 82.283 ± 2.336 |
| 0.6 | 90.170 ± 1.852 | 89.077 ± 1.852 | 88.036 ± 1.943 | 85.067 ± 2.124 | 81.585 ± 2.264 | 79.443 ± 2.321 | 84.474 ± 2.194 |
| 0.8 | 90.981 ± 1.737 | 90.127 ± 1.732 | 89.185 ± 1.881 | 86.562 ± 2.053 | 83.281 ± 2.128 | 81.293 ± 2.266 | 85.400 ± 2.053 |
| 1.0 | 91.415 ± 1.745 | 90.697 ± 1.694 | 89.849 ± 1.804 | 87.463 ± 2.015 | 84.339 ± 2.210 | 82.412 ± 2.172 | 85.844 ± 2.090 |
| 1.2 | 91.657 ± 1.692 | 91.089 ± 1.653 | 90.576 ± 1.767 | 89.089 ± 1.895 | 86.603 ± 2.078 | 85.111 ± 2.154 | 86.088 ± 2.145 |
| 2.0 | 92.146 ± 1.616 | 91.751 ± 1.686 | 91.428 ± 1.663 | 90.328 ± 1.796 | 88.485 ± 1.956 | 87.211 ± 2.098 | 85.754 ± 2.149 |
| 3.0 | 92.285 ± 1.651 | 91.964 ± 1.601 | 91.718 ± 1.651 | 91.034 ± 1.757 | 89.755 ± 1.855 | 88.989 ± 1.876 | 84.890 ± 2.276 |
| 4.0 | 92.278 ± 1.662 | 91.929 ± 1.679 | 91.785 ± 1.722 | 91.363 ± 1.701 | 90.404 ± 1.804 | 89.876 ± 1.829 | 84.246 ± 2.288 |
| 5.0 | 92.182 ± 1.667 | 91.922 ± 1.652 | 91.730 ± 1.719 | 91.492 ± 1.683 | 90.832 ± 1.729 | 90.358 ± 1.742 | 83.619 ± 2.390 |

Table 34: Results for the Wine dataset.

| BR | RF(nt_500) | RF(qs_ent) | RF(mn_4) | RF(mn_8) | RF(ml_4) | RF(ml_5) | RF(nf_all) |
|---|---|---|---|---|---|---|---|
| 0.2 | 97.548 ± 1.432 | 96.649 ± 1.856 | 96.674 ± 1.847 | 96.444 ± 1.891 | 96.413 ± 1.915 | 96.435 ± 1.965 | 95.716 ± 2.252 |
| 0.4 | 97.632 ± 1.363 | 97.067 ± 1.751 | 97.034 ± 1.743 | 96.840 ± 1.767 | 96.489 ± 1.785 | 96.447 ± 1.870 | 95.309 ± 2.617 |
| 0.6 | 97.694 ± 1.358 | 97.346 ± 1.619 | 97.298 ± 1.603 | 97.191 ± 1.642 | 96.677 ± 1.761 | 96.581 ± 1.755 | 95.073 ± 2.973 |
| 0.8 | 97.728 ± 1.347 | 97.368 ± 1.575 | 97.424 ± 1.519 | 97.331 ± 1.563 | 96.691 ± 1.697 | 96.579 ± 1.758 | 95.132 ± 3.017 |
| 1.0 | 97.764 ± 1.381 | 97.396 ± 1.578 | 97.455 ± 1.534 | 97.357 ± 1.560 | 96.728 ± 1.773 | 96.581 ± 1.807 | 94.927 ± 3.137 |
| 1.2 | 97.809 ± 1.312 | 97.416 ± 1.630 | 97.441 ± 1.528 | 97.385 ± 1.601 | 96.969 ± 1.638 | 96.812 ± 1.721 | 94.916 ± 3.081 |
| 2.0 | 97.761 ± 1.340 | 97.416 ± 1.663 | 97.567 ± 1.513 | 97.455 ± 1.559 | 97.039 ± 1.675 | 96.938 ± 1.789 | 94.067 ± 3.291 |
| 3.0 | 97.654 ± 1.465 | 97.362 ± 1.615 | 97.489 ± 1.541 | 97.452 ± 1.581 | 97.211 ± 1.682 | 97.051 ± 1.713 | 93.492 ± 3.291 |
| 4.0 | 97.649 ± 1.445 | 97.357 ± 1.639 | 97.500 ± 1.509 | 97.511 ± 1.513 | 97.334 ± 1.598 | 97.272 ± 1.595 | 93.101 ± 3.373 |
| 5.0 | 97.640 ± 1.421 | 97.329 ± 1.648 | 97.492 ± 1.536 | 97.438 ± 1.560 | 97.289 ± 1.612 | 97.236 ± 1.604 | 92.812 ± 3.454 |

Table 35: Results for the Ringnorm dataset.

| BR | RF(nt_500) | RF(qs_ent) | RF(mn_4) | RF(mn_8) | RF(ml_4) | RF(ml_5) | RF(nf_all) |
|-----|-----|-----|-----|-----|-----|-----|-----|
| 0.2 | 89.752 ± 2.383 | 88.622 ± 2.779 | 89.057 ± 2.718 | 90.107 ± 2.605 | 83.540 ± 3.072 | 81.078 ± 3.104 | 81.605 ± 3.569 |
| 0.4 | 92.597 ± 2.220 | 91.437 ± 2.413 | 91.802 ± 2.314 | 92.443 ± 2.212 | 89.123 ± 2.691 | 87.778 ± 2.827 | 86.065 ± 3.641 |
| 0.6 | 92.717 ± 2.390 | 91.538 ± 2.526 | 91.962 ± 2.455 | 92.335 ± 2.352 | 89.738 ± 2.701 | 88.995 ± 2.848 | 86.747 ± 3.468 |
| 0.8 | 92.370 ± 2.473 | 91.515 ± 2.582 | 92.060 ± 2.533 | 92.192 ± 2.413 | 89.898 ± 2.675 | 89.288 ± 2.742 | 86.725 ± 3.669 |
| 1.0 | 92.173 ± 2.457 | 91.325 ± 2.692 | 91.710 ± 2.543 | 91.957 ± 2.376 | 89.828 ± 2.619 | 89.232 ± 2.687 | 86.767 ± 3.679 |
| 1.2 | 91.977 ± 2.513 | 91.010 ± 2.693 | 91.407 ± 2.617 | 91.657 ± 2.539 | 90.160 ± 2.638 | 89.562 ± 2.832 | 86.563 ± 3.737 |
| 2.0 | 91.322 ± 2.523 | 90.442 ± 2.754 | 91.068 ± 2.593 | 91.213 ± 2.590 | 89.865 ± 2.754 | 89.453 ± 2.837 | 85.672 ± 3.747 |
| 3.0 | 90.670 ± 2.640 | 89.985 ± 2.752 | 90.365 ± 2.661 | 90.450 ± 2.597 | 89.757 ± 2.755 | 89.432 ± 2.788 | 84.672 ± 4.002 |
| 4.0 | 90.398 ± 2.648 | 89.603 ± 2.710 | 90.092 ± 2.730 | 90.173 ± 2.691 | 89.680 ± 2.698 | 89.405 ± 2.692 | 84.085 ± 3.918 |
| 5.0 | 90.173 ± 2.731 | 89.312 ± 2.864 | 89.882 ± 2.666 | 89.953 ± 2.717 | 89.555 ± 2.733 | 89.473 ± 2.724 | 83.438 ± 3.954 |

Table 36: Results for the Threenorm dataset.

| BR | RF(nt_500) | RF(qs_ent) | RF(mn_4) | RF(mn_8) | RF(ml_4) | RF(ml_5) | RF(nf_all) |
|-----|-----|-----|-----|-----|-----|-----|-----|
| 0.2 | 79.700 ± 2.653 | 77.880 ± 2.842 | 77.762 ± 2.973 | 77.563 ± 2.958 | 77.197 ± 2.973 | 76.840 ± 2.982 | 76.652 ± 3.294 |
| 0.4 | 80.050 ± 2.774 | 78.720 ± 2.921 | 78.812 ± 3.059 | 78.557 ± 2.957 | 78.468 ± 3.065 | 78.128 ± 3.138 | 77.513 ± 3.275 |
| 0.6 | 79.888 ± 2.623 | 79.040 ± 2.823 | 79.040 ± 2.753 | 78.963 ± 2.746 | 78.587 ± 2.734 | 78.342 ± 2.820 | 77.258 ± 3.135 |
| 0.8 | 79.852 ± 2.690 | 78.910 ± 2.977 | 79.252 ± 2.782 | 79.035 ± 2.856 | 78.733 ± 2.997 | 78.667 ± 2.891 | 77.218 ± 3.254 |
| 1.0 | 79.743 ± 2.617 | 78.723 ± 2.892 | 78.977 ± 2.792 | 79.025 ± 2.736 | 78.643 ± 2.856 | 78.505 ± 2.890 | 77.008 ± 3.304 |
| 1.2 | 79.725 ± 2.693 | 78.715 ± 2.903 | 78.967 ± 2.861 | 78.958 ± 2.820 | 78.868 ± 2.887 | 78.702 ± 2.938 | 76.598 ± 3.262 |
| 2.0 | 79.465 ± 2.681 | 78.663 ± 2.941 | 78.810 ± 2.659 | 78.925 ± 2.822 | 78.687 ± 2.827 | 78.528 ± 2.833 | 75.537 ± 3.555 |
| 3.0 | 79.032 ± 2.749 | 78.307 ± 2.811 | 78.503 ± 2.890 | 78.525 ± 2.860 | 78.390 ± 2.811 | 78.465 ± 2.860 | 74.313 ± 3.647 |
| 4.0 | 78.843 ± 2.693 | 78.053 ± 2.910 | 78.422 ± 2.954 | 78.313 ± 2.865 | 78.407 ± 2.845 | 78.373 ± 2.859 | 73.457 ± 3.735 |
| 5.0 | 78.810 ± 2.787 | 77.863 ± 3.004 | 78.178 ± 2.904 | 78.312 ± 2.964 | 78.258 ± 2.877 | 78.285 ± 2.944 | 72.995 ± 3.860 |

Table 37: Results for the Twonorm dataset.

| BR | RF(nt_500) | RF(qs_ent) | RF(mn_4) | RF(mn_8) | RF(ml_4) | RF(ml_5) | RF(nf_all) |
|-----|-----|-----|-----|-----|-----|-----|-----|
| 0.2 | 96.002 ± 1.397 | 94.825 ± 1.692 | 95.023 ± 1.634 | 94.927 ± 1.780 | 94.755 ± 1.720 | 94.590 ± 1.751 | 93.297 ± 2.170 |
| 0.4 | 95.688 ± 1.426 | 94.630 ± 1.742 | 94.977 ± 1.651 | 94.730 ± 1.676 | 94.363 ± 1.747 | 94.192 ± 1.831 | 92.850 ± 2.546 |
| 0.6 | 95.413 ± 1.533 | 94.360 ± 1.774 | 94.805 ± 1.646 | 94.540 ± 1.722 | 94.070 ± 1.728 | 93.942 ± 1.788 | 91.957 ± 2.703 |
| 0.8 | 95.285 ± 1.560 | 94.297 ± 1.800 | 94.567 ± 1.745 | 94.467 ± 1.717 | 93.993 ± 1.833 | 93.722 ± 1.836 | 91.260 ± 2.948 |
| 1.0 | 95.093 ± 1.594 | 93.975 ± 1.833 | 94.448 ± 1.804 | 94.253 ± 1.723 | 93.770 ± 1.740 | 93.625 ± 1.911 | 90.832 ± 3.036 |
| 1.2 | 95.008 ± 1.634 | 93.957 ± 1.825 | 94.355 ± 1.727 | 94.288 ± 1.867 | 93.797 ± 1.901 | 93.678 ± 1.893 | 90.383 ± 3.141 |
| 2.0 | 94.617 ± 1.728 | 93.653 ± 2.006 | 94.123 ± 1.838 | 94.038 ± 1.772 | 93.542 ± 2.007 | 93.307 ± 1.991 | 88.787 ± 3.731 |
| 3.0 | 94.413 ± 1.814 | 93.378 ± 1.957 | 93.923 ± 1.828 | 93.955 ± 1.892 | 93.438 ± 1.949 | 93.322 ± 2.010 | 87.398 ± 3.882 |
| 4.0 | 94.197 ± 1.846 | 93.200 ± 1.975 | 93.628 ± 1.848 | 93.807 ± 1.884 | 93.425 ± 1.977 | 93.302 ± 2.006 | 86.270 ± 4.283 |
| 5.0 | 94.125 ± 1.866 | 93.057 ± 1.966 | 93.680 ± 1.917 | 93.647 ± 1.897 | 93.435 ± 1.907 | 93.342 ± 1.969 | 85.280 ± 4.206 |

Table 38: Results for the Waveform dataset.

| BR | RF(nt_500) | RF(qs_ent) | RF(mn_4) | RF(mn_8) | RF(ml_4) | RF(ml_5) | RF(nf_all) |
|-----|-----|-----|-----|-----|-----|-----|-----|
| 0.2 | 86.165 ± 2.424 | 84.730 ± 2.706 | 84.752 ± 2.666 | 84.630 ± 2.705 | 84.067 ± 2.777 | 83.598 ± 2.739 | 83.990 ± 2.788 |
| 0.4 | 85.635 ± 2.441 | 84.913 ± 2.489 | 84.702 ± 2.578 | 84.658 ± 2.591 | 84.900 ± 2.591 | 84.465 ± 2.553 | 82.855 ± 3.023 |
| 0.6 | 85.328 ± 2.322 | 84.747 ± 2.563 | 84.375 ± 2.433 | 84.313 ± 2.561 | 84.857 ± 2.639 | 84.725 ± 2.619 | 81.982 ± 3.297 |
| 0.8 | 84.868 ± 2.409 | 84.538 ± 2.441 | 84.253 ± 2.667 | 84.253 ± 2.515 | 84.823 ± 2.576 | 84.827 ± 2.616 | 81.252 ± 3.185 |
| 1.0 | 84.682 ± 2.396 | 84.367 ± 2.447 | 83.973 ± 2.479 | 83.970 ± 2.690 | 84.660 ± 2.573 | 84.632 ± 2.539 | 80.578 ± 3.406 |
| 1.2 | 84.435 ± 2.406 | 84.135 ± 2.592 | 83.998 ± 2.537 | 83.910 ± 2.563 | 84.552 ± 2.562 | 84.598 ± 2.650 | 80.015 ± 3.251 |
| 2.0 | 83.948 ± 2.494 | 83.758 ± 2.533 | 83.332 ± 2.500 | 83.327 ± 2.731 | 84.000 ± 2.582 | 84.058 ± 2.684 | 78.445 ± 3.620 |
| 3.0 | 83.585 ± 2.478 | 83.368 ± 2.600 | 83.293 ± 2.563 | 83.230 ± 2.704 | 83.788 ± 2.702 | 83.740 ± 2.625 | 77.073 ± 3.826 |
| 4.0 | 83.485 ± 2.539 | 83.398 ± 2.528 | 83.177 ± 2.706 | 83.073 ± 2.593 | 83.507 ± 2.534 | 83.590 ± 2.503 | 76.318 ± 3.853 |
| 5.0 | 83.437 ± 2.509 | 83.282 ± 2.544 | 83.083 ± 2.552 | 83.055 ± 2.503 | 83.230 ± 2.609 | 83.468 ± 2.596 | 75.653 ± 3.977 |

Table 39: Results for the LED Display Domain dataset.

| BR | RF(nt_500) | RF(qs_ent) | RF(mn_4) | RF(mn_8) | RF(ml_4) | RF(ml_5) | RF(nf_all) |
|----|-----------|-----------|----------|----------|----------|----------|-----------|
| 0.2 | 61.648 ± 3.594 | 57.242 ± 4.151 | 58.780 ± 4.167 | 61.237 ± 3.791 | 57.737 ± 4.245 | 54.337 ± 4.506 | 63.862 ± 3.841 |
| 0.4 | 63.392 ± 3.392 | 61.350 ± 3.712 | 62.823 ± 3.885 | 65.780 ± 3.654 | 64.990 ± 3.818 | 64.850 ± 3.512 | 64.633 ± 3.597 |
| 0.6 | 63.365 ± 3.434 | 61.950 ± 3.708 | 62.835 ± 3.763 | 65.375 ± 3.708 | 65.847 ± 3.627 | 66.018 ± 3.704 | 62.800 ± 3.736 |
| 0.8 | 63.285 ± 3.478 | 61.343 ± 3.613 | 62.977 ± 3.735 | 64.765 ± 3.697 | 65.703 ± 3.673 | 66.425 ± 3.750 | 61.330 ± 3.704 |
| 1.0 | 63.025 ± 3.481 | 61.593 ± 3.608 | 63.047 ± 3.787 | 64.528 ± 3.715 | 65.448 ± 3.790 | 66.590 ± 3.755 | 60.175 ± 3.837 |
| 1.2 | 62.818 ± 3.516 | 61.617 ± 3.801 | 62.740 ± 3.485 | 64.327 ± 3.701 | 64.993 ± 3.476 | 65.320 ± 3.472 | 59.453 ± 3.951 |
| 2.0 | 62.553 ± 3.622 | 61.428 ± 3.795 | 62.062 ± 3.696 | 63.653 ± 3.511 | 64.565 ± 3.620 | 65.050 ± 3.650 | 57.205 ± 4.111 |
| 3.0 | 62.110 ± 3.544 | 60.918 ± 3.705 | 61.262 ± 3.575 | 62.790 ± 3.642 | 63.873 ± 3.489 | 64.218 ± 3.450 | 55.770 ± 4.185 |
| 4.0 | 61.888 ± 3.528 | 61.000 ± 3.822 | 61.155 ± 3.752 | 62.073 ± 3.700 | 63.287 ± 3.582 | 63.932 ± 3.513 | 55.203 ± 4.229 |
| 5.0 | 61.848 ± 3.570 | 61.240 ± 3.724 | 60.912 ± 3.687 | 61.622 ± 3.713 | 62.742 ± 3.714 | 63.420 ± 3.666 | 54.860 ± 4.338 |

# C   BOOTSTRAP RATE CURVES

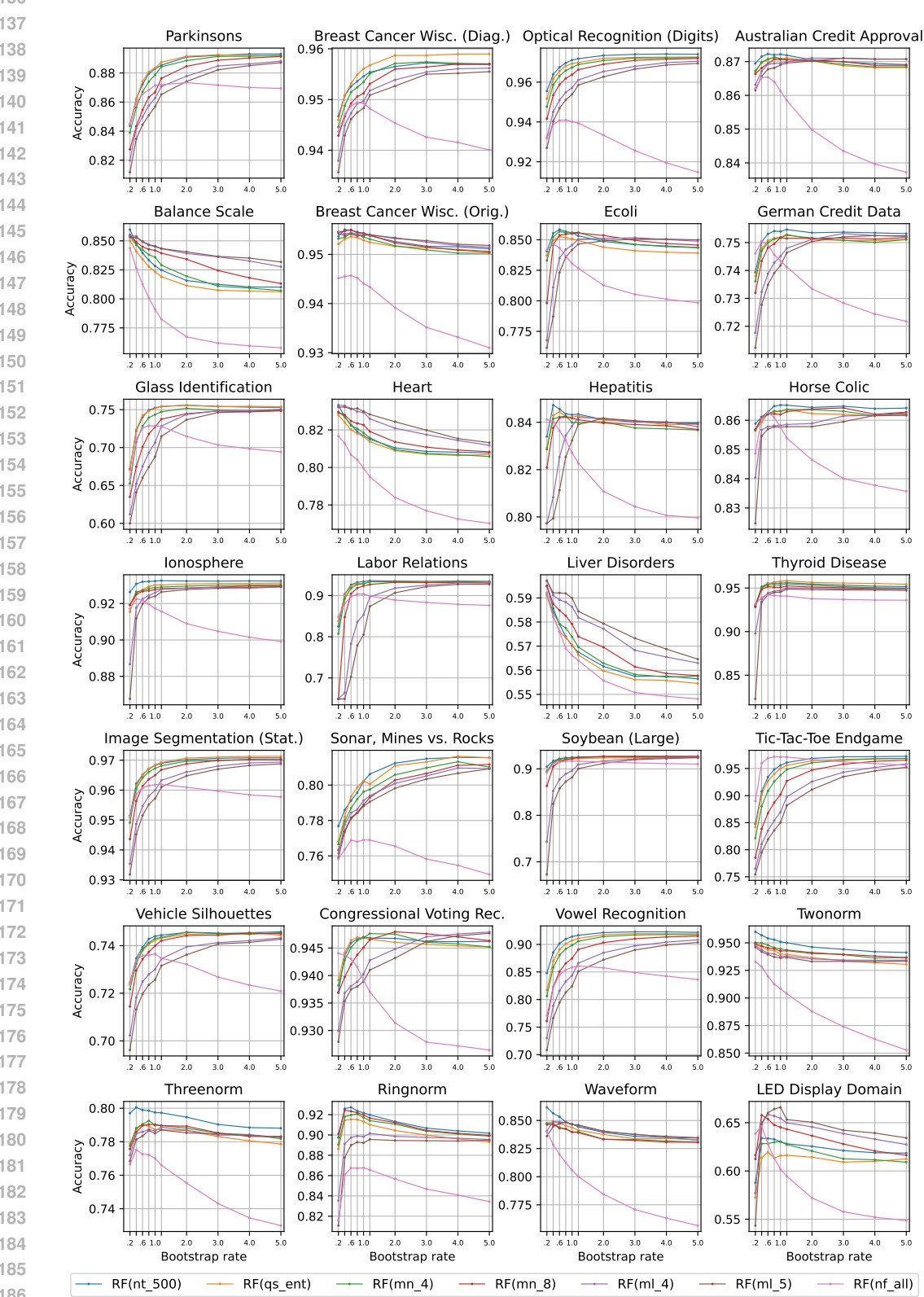

Figure 4: Characteristics of bootstrap rate curves for datasets not shown in Fig. 2.

