# OpenReview forum: "Bootstrap Sampling Rate Greater than 1.0 May Improve Random Forest Performance"
_ICLR.cc/2025/Conference — Submitted to ICLR 2025_

### Official Review · Reviewer_ocp7 · 2024-10-19

**Soundness:** 2
**Presentation:** 2
**Contribution:** 1
**Rating:** 3
**Confidence:** 4

**Summary:**

The paper studies the effects of the Bootstrap Rate (BR) parameter on the test set accuracy of random forest classifiers. The authors claim that it is possible to obtain improvements (not always) by setting BR > 1, which is rarely done in practice. The authors develop a classifier to predict if the optimal BR is larger than 1 based on the dataset.

**Strengths:**

* Good scope of study in terms of datasets and covering a wide range of hyperparameters
* Exploration of the effects of a bootstrap rate that is generally understudied
* The idea of determining the BR based on data statistics is interesting

**Weaknesses:**

* The analysis carried out in table 1 is not explained in a clear enough way. For instance, the paired t-test is not fully described and is hard to understand. For a result listed as a key contribution in the paper, the procedure should be explicitly described.
* Figure two does not include error bars, and in almost all cases the effect of the BR parameter seems very small in terms of accuracy.
* The classifier only predicts if the optimal BR (a term that need to be formally defined) is larger than 1. It would be more useful for practitioners to predict the actual rate.
* Generally I feel as if the findings in this study align well with the hypothesis that BR > 1 is as good, but not better, than BR=1. When making a claim that the optimal BR is equal to a certain value, the authors rely on finite sample analysis, which might be susceptible to random fluctuations.

**Questions:**

* By "2-fold stratified cross-validation, repeated 200 times, was applied, yielding 400 results", do the authors mean that they did a 50/50 (train/test) split and ran two experiments for each configuration?
* Can the authors please fully describe their hypothesis testing procedure carried out in Table 1?
* Can the authors please formally define the optimal BR?

---

> ### Author Response · Authors · 2024-11-23
>
> Thank you for your constructive feedback. Our responses to the points you raised are provided below.
>
> *W1*
>
> We will clarify and expand the description provided in the "Statistical significance" paragraph of Section 4 in the final version of the paper. The main focus of the experiments is to examine whether (and, if so, how often) RF with a hyperparameter value of BR > 1 can yield better results than RF with the standard value of BR ≤ 1.
> To achieve this, all 180 RF configurations were divided into two groups: those with BR ≤ 1 (the first group, 90 configurations) and those with BR > 1 (the second group, 90 configurations). We then examined the accuracy achieved by the best configuration in each group and compared them. For 20 out of 36 datasets, the best-performing configuration had BR > 1.
> Next, we aimed to assess the statistical significance of these results. To this end, we compared the results of the best configuration with those of all configurations in the other group (i.e., if the best model had BR ≤ 1, we compared it to all configurations with BR > 1, and vice versa) using a paired t-test and reported the maximum p-value. Each realization of the t-test involved providing pairs of results corresponding to 400 train-test splits in the cross-validation process.
> Finally, for several significance levels, we considered only those results for which the p-value was smaller than the specified significance level. In approximately half of the cases, BR > 1 produced statistically better results than BR ≤ 1 (depending on the chosen significance level, this is sometimes slightly less or slightly more than half).
>
> *W2*
>
> Creating 7 bars with error/standard deviation for the same values on the x-axis (BR) would result in poor readability. The differences in achieved accuracy for different BR values are not minor, as demonstrated by the conclusive results of statistical tests. A table presenting standard deviations for each dataset and RF configuration will be added to the final version of the paper.
>
> *W3*
>
> The optimal BR refers to the value of the BR hyperparameter in the RF configuration that achieves the highest average accuracy on a given dataset. In this study, we analyze 180 RF configurations: 10 BR values for each of the 18 hyperparameterizations (varying values of hyperparameters other than BR).
> Our main motivation was to demonstrate that statistics based on the number of neighbors adequately describe the problem (they are reasonable predictors): “To further assess how well the above set of attributes describes the problem, we used them (along with base k_l statistics) to build a binary classifier predicting whether the optimal BR across all RF configurations is ≤ 1 or > 1 for a given dataset.”. The goal was not to create a high-quality practical classifier.
> Predicting a specific BR value would be useful, though ambitious and challenging. The main issue we currently see is how to properly define the error function, given the uneven distribution of BR values. Looking at Fig. 2 and Fig. 4, we observe that the variability in the range [0.2, 1] is much larger than in the range [1, 5]. Using standard regression metrics like MSE or MAE to measure prediction quality wouldn't provide much insight. For example, a prediction error of 0.4 when the optimal BR is 0.6 generally leads to a much greater difference in model accuracy than the same error of 0.4 when the optimal BR is 0.2. An alternative approach would be to measure how much worse the accuracy of the model with the predicted BR is compared to the accuracy achieved by the optimal BR. Predicting the exact BR value is an avenue for future research.
>
> *W4*
>
> To verify whether the results are not coincidental, we performed statistical tests. These tests were conducted at various significance levels covering a wide range of values. In approximately half of the cases, BR > 1 produced statistically better results than BR ≤ 1 (depending on the chosen significance level, this is sometimes slightly less or slightly more than half). Moreover, in most cases, the selection of datasets was not arbitrary—30 out of 36 datasets were those used in the reference paper.
>
> *Q1*
>
> Yes, a 50%/50% train-test split was used, with two experiments conducted for each.
>
> *Q2*
>
> The answer is provided in W1. Please let us know if anything remains unclear.
>
> *Q3*
>
> The optimal BR refers to the value of the BR hyperparameter in the RF configuration that achieves the highest average accuracy on a given dataset. In this study, we analyze 180 RF configurations: 10 BR values for each of the 18 hyperparameterizations (varying values of hyperparameters other than BR).

---

> > ### Comment · Reviewer_ocp7 · 2024-11-23
> > **thank you for your response**
> >
> > I thank the authors for their response, I stand by my score.

---

### Official Review · Reviewer_zB9e · 2024-10-24

**Soundness:** 3
**Presentation:** 3
**Contribution:** 1
**Rating:** 1
**Confidence:** 5

**Summary:**

This article investigates the impact of the bootstrap rate (BR) on the predictive performances of random forests. BR is a hyperparameter of random forests: each tree of a random forests are built by drawing with replacement BR*n observations among the n original ones.
Previous works have tested the impact of BR values ranging from 0.2 to 1.2. In this paper, the authors compare the performance of various RF configurations (varying the number of trees, maximal depth, number of features randomly selected in each cell...) together with different BR values ranging from 0.2 to 5.

Experimental results over 36 open source data sets show that a BR larger than one is associated to good predictive performances for most data sets. The authors explain these results by introducing a statistics measure over a data set, corresponding to the inhomogeneity of the data set. More precisely, the parameter $k\_l$ stands for the number of observations in the data set that have $l$ observations with the given label among their $k$ nearest neighbors. The authors demonstrate that the optimal BR is correlated to this measure.

Finally, the authors also introduce two binary classifiers that predict if a given data set has an optimal BR greater or lower than 1. These classifiers are respectively trained on 36 and 24 data sets.

**Strengths:**

The paper studies the impact of BR on random forests performances and provide insights about the characteristics of data sets for which BR >1 should be preferred. Extensive simulations are done to corroborate the proposed statements.

**Weaknesses:**

- Several related works are missing. Besides, it is not clear from the related work section which contributions are theoretical or practical. See for example:
    - For empirical performances, see Section 4 in Random Forests for Big Data https://arxiv.org/abs/1511.08327
    - For theoretical results quantifying the uncertainty of random forests, depending on BR, see https://jmlr.org/papers/volume17/14-168/14-168.pdf or https://arxiv.org/pdf/1405.0352
    - In https://www.esaim-ps.org/articles/ps/pdf/2018/01/ps170099.pdf, the authors provide the expression of the optimal subsampling rate for median forests
    - Regarding Breiman's forests, the analysis of https://arxiv.org/pdf/2006.06998 takes into account bootstrap

- Experiments are done for classification data sets only. The paper would be strengthened if experiments were done also for regression problems.

I have several concerns related to the design of experiments, and more specifically with Table 1:
- No standard deviations are displayed. Thus, we are not sure if the best method is really the best or if many other RF configurations have performances close to the best one.
- Looking at Table 2 in Martinez-Munoz and Suarez (2010), which displays the average test error for different BR ratios, the standard deviations are quite important. For example, the standard deviation of the error (and thus the accuracy) is between 7 and 8 for the audio data set, whereas you display the accuracy with 3 decimals.
- Besides, the test set is used for two different objectives: selecting the best RF configuration and computing its error. Thus, the error of the best RF configuration is rather optimistic. It would be more sound to evaluate the performance on a different data set (or to use the out-of-bag error of the forest). Thus, t-test results do not seem conclusive. Besides, as you consider several t-test, you should probably correct the significance level $\alpha$ accordingly, taking into account multiple testing issues.
- Given the performances displayed in Table 1, I would like to see the same experiment run with a Baseline RF whose number of trees is set to 500. Indeed, this parameter seems to be the most important one, whereas in practice it is fairly easy to tune (the higher the better in terms of accuracy).
- A Table with all values of RF configuration / BR rate with standard deviations would give more precise information.
- l.335 The fact that optimal performances are reached for two very different values of BR (0.2 and 5) by slightly changing the data set is worrying. One reason can be that the two values are good for the two data sets, which is in contradiction with your study, showing that BR >1 is better than the opposite. This is why metrics for all configurations with standard deviations should be displayed.

**Questions:**

- l.47 "we expect 36.8% of observations to be absent in each bootstrap sample." We know that the probability for one observation to be absent of a bootstrap sample is 0.368. However, I am not sure that this corresponds to the percentage of observations absent from the bootstrap sample, as the events $B_i$ ("presence of the $i$th observation in the bootstrap sample") are not independent. Can you prove the statement or give a reference?
- Looking at the paper of Martinez-Munoz and Suarez (2010), it appears that they do not use split randomization in their random forests implementation. This could explain the difference between their results and the results obtained in the present paper. This should be clearly mentioned somewhere.
- l.98 what is the rationale behind standardizing one-hot encoded features? Tree based methods do not usually need input standardization to work.
- The ranges of values for the different hyperparameters are not consistent: the max depth varies between 10 and 25 (which results in leaf containing between n/2^10 and n/2^25 observations if the tree is balanced) while the minimum number of observations per leaf varies between 2 and 5. These ranges of values should be harmonized.
- A twofold cross-validation is performed 200 times. Usually, a 5-fold cross validation is performed. Is there any reason for choosing this value? Can you display standard deviation averaged across runs?
- A BR value much larger than one would lead to use almost all observations for each tree construction. This would probably damage the computation of the out-of-bag error, which is an important feature of random forests, as it allows us not to divide the data set into a train and a test set to compute the RF error. Could you study how the out-of-bag error is influenced by the BR value?
- l.206 "Finally, BR= 1, defined in the original formulation of the bootstrapping procedure and the most frequently used
value, performed relatively poorly." This is not shown by the experiments: the only things shown here with respect to BR=1 is that it is often not the best configuration. But it can be the second-best most of the time.
- l.340 notation $k\_l$ is uncommon and misleading as it counts a number of observations. I would rename it $n_{k,l}$ for clarity.
- Table 2. What is the rationale for studying very low values of $k$ and $l$ with respect to the data set size? Choosing a fix number of nearest neighbors $k$ for all data sets corresponds to considering neighborhood of different sizes across data sets. Thus, I wonder if such a measure $k\_l$ corresponds to a characteristic of the data set.
- l.431 Regarding the bootstrap rate prediction, I am not sure to understand the final methodology. More precisely, I do not understand how features are selected: "based on the absolute value of the Spearman rank-order coefficient, calculated separately for each run on the training instances". Does it mean that, at first, 12,000 features are computed, then the correlation with the best BR is computed, and only the ten best are kept? Since computing the best BR requires a validation set, performances should be computed on a test set. Besides, does the procedure require running random forest with all BR values (in order to compute the best ten values among the 12,000 original features) in order to build a classifier to choose the best BR value?
- The resulting classifier predict whether the best BR is larger than one. More practically, I would be more interested in estimating the best BR. How does your analysis help the practitioners?

---

> ### Author Response · Authors · 2024-11-23
>
> Thank you for your careful review and helpful suggestions. Please see our responses to the comments below.
>
> *Weaknesses*
>
> “Several related works are missing …” The paper "Impact of subsampling and tree depth on random forests" has been cited. The other three papers do not directly refer to the topic of the impact of BR values on model quality. We will add them in a more general context in the related literature discussion.
>
> “Experiments are done for classification data sets only …”
> You are absolutely right; this is on our list of future research topics. However, it requires considerable work involving data preprocessing and modeling for several dozen additional datasets and extends beyond the scope of this paper.
>
> “No standard deviations are displayed. Thus, we are not sure …”
> A table presenting standard deviations for each dataset and RF configuration will be added to the final version of the paper. Please observe, however, that what answers the question of whether and to what extent a given model is the best is the p-value analysis of statistical tests, which we have conducted.
>
> “Looking at Table 2 in Martinez-Munoz and Suarez …”
> In the paper, we demonstrate that BR > 1 is often better than BR ≤ 1. To make this claim, we need to perform a statistical test comparing the best configuration with BR > 1 to the best configuration with BR ≤ 1. Conducting a paired Student's t-test requires the individual results from each experiment (400 pairs). Focusing solely on standard deviation would not provide more insight than the statistical tests and, in fact, would be less informative.
>
> “Besides, the test set is used for two different objectives …”
> We followed the standard procedure for splitting the data into training and test sets, the same as in the work by Martinez-Munoz and Suarez (2010). What we actually need to determine is:
> - When the globally best configuration had BR ≤ 1: Identify the configuration with BR ≤ 1 that, when compared to all configurations with BR > 1, minimizes the p-value across all comparisons.
> - When the globally best configuration had BR > 1: Identify the configuration with BR > 1 that, when compared to all configurations with BR ≤ 1, minimizes the p-value across all comparisons.
>
> We assumed that the configuration minimizing the p-value would also be the global winner. After manually verifying a few datasets, this assumption holds true. However, theoretically, a situation could occur where a configuration that does not have the highest global accuracy minimizes the p-value because it has a smaller standard deviation (i.e., it is more stable) than the global winner.
> Therefore, to ensure full accuracy, in the final version of the paper, we will conduct tests for every pair of models with BR ≤ 1 and BR > 1. With this approach, there is no need to select the best configuration, and thus, no additional datasets are required.
>
> “Besides, as you consider several t-test …”
> For each dataset, the minimum p-value was calculated. We demonstrated that, across a wide range of significance levels, the number of datasets for which the best model has BR ≤ 1 and the number of datasets for which the best model has BR > 1 are similar. Based on the calculated p-values, the reader can apply a correction, the choice of which is not straightforward and depends on the desired level of conservativeness.
>
> "Given the performances displayed in Table 1 … "
> As you have noted, in general, the more trees, the better the accuracy. This raises the question of why the number of trees has to be set to 500 and not, for example, 200, 800, or 1000. We adopted 100 as the number of trees because it is the default value or greater than the default in most libraries. For instance, the default values for the number of trees in RFs in popular packages like scikit-learn, Weka, and H2O are 100, 100, and 50, respectively.
>
> “A Table with all values of RF configuration / BR rate …”
> We will add it to the final version of the paper.
>
> “l.335 The fact that optimal performances … “
> Such a situation does not occur here. In the first case, the curve is increasing, while in the second case, it is (roughly) decreasing across all BR values. We will add the plots to the final version of the paper.

---

> ### Author Response · Authors · 2024-11-23
>
> *Questions*
>
> “l.47 "we expect 36.8% of observations …”
> Efron, Bradley, and Robert Tibshirani. "Improvements on cross-validation: the 632+ bootstrap method." Journal of the American Statistical Association 92.438 (1997): 548-560.
>
> "Looking at the paper of Martinez-Munoz and Suarez (2010) … "
> The reasons for the differences are provided in the second paragraph of the conclusion. We will supplement them with the suggested argument.
>
> “l.98 what is the rationale behind standardizing … “
> You are right. It was unnecessary and did not affect the results at all.
>
> “The ranges of values for the different hyperparameters are not consistent …”
> - The provided values refer to the scenario of perfectly balanced trees. In practice, trees are typically unbalanced, and the paths from root to leaves vary in length.
> - I am not certain why the sizes of trees for this pair of parameters should be harmonized. Three out of the six hyperparameters we test (number of trees, splitting criterion, and number of features) impose no restrictions on tree structure. Moreover, greater diversity (within reasonable limits) among trees increases the chances of finding an even better solution.
>
> “A twofold cross-validation is performed 200 times … ”
> Two-fold cross-validation results in a larger test set compared to five-fold cross-validation. This makes the results on the test set more stable. As a result, with a given number of repetitions, more outcomes are conclusive at the specified significance level (for a larger number of datasets, the p-value from the test is smaller than the specified significance threshold).
> A table presenting standard deviations for each dataset and RF configuration will be added to the final version of the paper.
>
> “A BR value much larger than one would lead to …”
> Indeed, the number of observations in the OOB set for BR > 1 decreases rapidly, making the set highly questionable for use. For BR = 2, BR = 3, BR = 4, and BR = 5, the proportion of observations in the OOB set for large n converges to approximately 13.53%, 4.98%, 1.83%, and 0.67% of the total number, respectively.
>
> “l.206 "Finally, BR= 1, defined in the original formulation, …”
> We will calculate the average rank of each BR and include it in the final version of the paper.
>
> “l.340 notation kl is uncommon and misleading …”
> These values are normalized to 100 as per: 'Then, we performed normalization so that for each k, the sum k_0, ..., k_k equaled 100, making those values comparable between datasets of varying sizes.' Thus, these are floating-point numbers.
>
> “Table 2. What is the rationale for studying very low values of k and l … ”
> - As mentioned above, these values are normalized to 100 and represent the percentage of a particular number of neighbors.
> - The considerations regarding k_l pertain to how the leaves are constructed. For most configurations, the size of the leaves (i.e., the number of instances on which a leaf is based) is independent of the size of the dataset.
>
> “l.431 Regarding the bootstrap rate prediction …”
> 12 685 features were created. Then, for each experiment within cross-validation:
> - We calculate the 10 most correlated with the target features based solely on the training set.
> - Next, we train each of the 180 random forest parameterizations using the top k most correlated features (k = 1, ... 10).
> - We evaluate each of the above 1800 configurations on the test set.
>
> “Besides, does the procedure require running random forest with all BR values …”
> No, selecting the 10 most correlated features required calculating correlations (separately for each split in cross-validation), not training a random forest.

---

> ### Author Response · Authors · 2024-11-23
>
> Cont.
> “The resulting classifier predict whether the best BR is larger than one …”
> Our main motivation was to demonstrate that statistics based on the number of neighbors adequately describe the problem (they are reasonable predictors): “To further assess how well the above set of attributes describes the problem, we used them (along with base k_l statistics) to build a binary classifier predicting whether the optimal BR across all RF configurations is ≤ 1 or > 1 for a given dataset.”. The goal was not to create a high-quality practical classifier.
>
> Predicting a specific BR value would be useful, though ambitious and challenging. The main issue we currently see is how to properly define the error function, given the uneven distribution of BR values. Looking at Fig. 2 and Fig. 4, we observe that the variability in the range [0.2, 1] is much larger than in the range [1, 5]. Using standard regression metrics like MSE or MAE to measure prediction quality wouldn't provide much insight. For example, a prediction error of 0.4 when the optimal BR is 0.6 generally leads to a much greater difference in model accuracy than the same error of 0.4 when the optimal BR is 0.2. An alternative approach would be to measure how much worse the accuracy of the model with the predicted BR is compared to the accuracy achieved by the optimal BR. Predicting the exact BR value is an avenue for future research.

---

> > ### Comment · Reviewer_zB9e · 2024-11-25
> >
> > Thank you for your response! I still have the following concerns:
> > - your analysis in Table 1 compares the best configuration among RF with $BR \leq 1$ with the best configuration among RF with $BR > 1$. Based on Table 1, it is impossible to see if the other BR values among a group (for example, in the group $BR \leq 1$) are close to the optimal one. Displaying the best BR value is somehow misleading, as there can be a large variability among the BR values of a same group, and thus several BR values can be close to optimality. This is why I advocate for displaying standard deviations for all BR values.
> > - Even if I understand the motivation of designing the so-called $kl$ statistics, and the fact that they are somehow related to the optimal BR, I fail to see the practical relevance of such a study, since, as stated by the authors, the goal is not to predict the optimal BR.
> > All in all, I find the research topic interesting, but I am not convinced by the relevance/applicability of the empirical results presented in the paper.  For these reasons, I will keep my score unchanged.

---

### Official Review · Reviewer_Lo5U · 2024-11-01

**Soundness:** 2
**Presentation:** 2
**Contribution:** 2
**Rating:** 5
**Confidence:** 3

**Summary:**

The paper explores the use of bootstrap sampling in Random Forests, specifically focusing on the bootstrap rate (BR), which is the ratio of the number of observations in each bootstrap sample to the total number of training instances. While traditional methods use a BR of 1 (sampling equal to the original dataset size), the authors investigate the effects of higher BR values, ranging from 1.2 to 5.0, on classification accuracy. Additionally, the authors create a binary classifier to predict whether the optimal BR for a given dataset is less than or equal to 1 or greater than 1.

**Strengths:**

1)	Overall, this work re-examines the bootstrap rate in random forests, considering a wide range of BR values and multiple datasets, which throw light on focusing some basic machine learning configs.

2)	Sufficient experiments on multiple datasets are carried out, which involves various aspect on BR values, such as the relationship between BR and classification accuracy, the influence of different RF configurations, and the dependence of the optimal BR on the dataset.

**Weaknesses:**

1)	Although different BR values have been extensively studied, the explanations for why certain datasets exhibit specific behaviors at specific BR values may not be deep and comprehensive enough, especially for some complex patterns and anomalies. For example, for a model like RF (nf all) that exhibits special behaviors, although some hypotheses have been proposed, more research may be needed to understand the exact mechanism behind it.

2)	When analyzing the relationship between the optimal BR value and the dataset, although some attributes and characteristics of the dataset are taken into account, there may be other undiscovered factors that affect the selection of the optimal BR value, which may play an important role in different datasets and application scenarios.

3)	The article does not analyze time performance related issues in detail. In practical applications, different BR values and model configurations may have a significant impact on training and prediction time, which may be a direction for future research.

**Questions:**

1)	Relationship between BR value and classification accuracy should be supported by more theoretical analysis and illustrated in a more rigorous form.

2)	It is presumed that “RF(nf all) will perform well on datasets with a high proportion of insignificant or less significant features, as it may avoid building trees primarily based on these features”. Could you provide some stronger evidence or more in-depth analysis?

---

> ### Author Response · Authors · 2024-11-23
>
> Thank you for your valuable feedback. Below are our responses to the comments and suggestions you provided.
>
> *W1*
>
> - Providing exact explanations for why certain datasets exhibit specific behaviors at particular BR values (i.e., detailed explanations of the BR curve shape) is likely impossible due to the following reasons:
>     - The problem is highly nuanced, as illustrated in Fig. 3.
>     - It depends on other RF parameters.
>     - In general, BR is a continuous variable, adding complexity to the analysis.
> This study is the first to analyze the factors influencing the shape of the BR curve. While we have provided some initial insights, we agree that further exploration of this topic is needed.
> - We proposed hypotheses to explain why RF(nf_all) behaves differently. A more detailed explanation lies beyond the scope of this study, as it pertains to a hyperparameter other than BR—specifically, the number of features considered at each node split.
>
> *W2*
>
> To the best of our knowledge, this is the first study to analyze the factors influencing the optimal BR. As such, the topic remains underexplored. We hope that further research will be conducted to gain a deeper understanding of this area.
>
> *W3*
>
> Naturally, this point is mentioned in the final paragraph of Section 4.
>
> *Q1*
>
> Definitely, that would be valuable. However, considering that the theoretical analysis of RFs is challenging due to their complexity (Duroux, Roxane and Scornet, Erwan. Impact of subsampling and tree depth on random forests. ESAIM: PS, 22:96–128, 2018.) and given the diversity of datasets, it is not a simple task. Nonetheless, it is certainly a promising avenue for further research.
>
> *Q2*
>
> An alternative to considering all features when splitting a node is randomly selecting log2 or sqrt (these are standard library values, which we also use) features and considering only those for the split. For example, for 25 features, log2 = 5 and sqrt = 5; for 100 features, log2 = 7 and sqrt = 10. Thus, only a small subset of features is considered.
> In situations where there are many irrelevant or weakly relevant features, it is quite common for the set of randomly selected features for a given node split to consist solely of weak features. As a result, the resulting split may not significantly improve the model's ability to generalize. Conversely, when all features are considered, there is always the possibility of selecting strong features. However, in the broader context, this approach is generally not advantageous, as it reduces the diversity of the trees.

---

### Official Review · Reviewer_GY4H · 2024-11-03

**Soundness:** 1
**Presentation:** 1
**Contribution:** 1
**Rating:** 1
**Confidence:** 5

**Summary:**

The paper investigates the impact of the bootstrap rate (BR) greater than 1.0 on Random Forest (RF) performance, concluding that higher BR values may lead to statistically significant improvements. The paper is purely empirical. I have a few concerns primarily regarding the lack of insight and the generalizability of the findings.

**Strengths:**

Extensive experimental studies

**Weaknesses:**

1. Lack of Insight: The study would greatly benefit from a discussion on the underlying rationale behind the observed improvements in classification accuracy with BR > 1. Although the (very limited) empirical results suggest that higher BR values could potentially improve performance, the paper does not offer insights into why this might be the case. Introducing a simplified, theoretical case, such as a linear Gaussian model, could help illustrate the mechanisms at play and provide a foundation for these empirical observations. This addition would make the findings more interpretable and provide value beyond experimental results.

2. Experimental Setup and Generalizability: While the authors employ a diverse dataset collection, the setup appears somewhat arbitrary, raising questions about the generalizability of the results. For example, RF performance is known to vary significantly with the number and complexity of categorical variables, which can greatly influence tree-based methods. Controlling for these factors or providing more context on dataset selection criteria would help clarify the scope of the findings and allow for a better assessment of their relevance to broader cases.

3. Classifier for BR Selection: The approach to developing a classifier to predict optimal BR usage lacks a solid rationale. The selection of 18 RF configurations and 10 BR values per dataset does not align with the principles of cross-validation, where data should be permutable to make the validation results meaningful/generalizable. This choice weakens the validity of the classification model and its potential for reliable application across diverse datasets.

**Questions:**

None

---

> ### Author Response · Authors · 2024-11-23
>
> Thank you for your thoughtful insights. Please find our responses to the comments below.
>
> *W1*
> - We are not sure why you consider the empirical results to be very limited. We conducted experiments on 36 diverse datasets, including 30 from the reference paper (Gonzalo Martınez-Munoz and Alberto Suarez. Out-of-bag estimation of the optimal sample size in bagging. Pattern Recognition, 43(1):143–152, 2010.) and 6 additional datasets introduced by us. Results achieved by models with BR ≤ 1 and BR > 1 were compared using statistical tests. Across all analyzed significance levels, the number of datasets where BR ≤ 1 outperformed BR > 1 and vice versa was very close.
> - In Section 5, "Towards Understanding the Optimal Bootstrap Rate," we analyze the factors that influence the optimal BR. First, we demonstrated that an approach considering high-level characteristics, such as the number of features (divided into binary and continuous) or the number of observations, is not effective. Similarly, taking the number of clusters into account does not yield meaningful results. Next, we showed that the problem is highly nuanced, and even minor differences in the data can significantly influence the shape of the BR curve. Analyzing the neighborhood structure of individual observations, which impacts the shape of the leaves in random forest trees, brought us closer to understanding the problem. The main observation is that the optimal BR value for heterogeneous datasets tends to be lower than for homogeneous datasets. We also demonstrated that variables constructed based on the neighborhood structure of individual instances are relatively highly correlated with the optimal BR value.
> - Could you please provide more details on your suggestion to introduce a linear Gaussian model? Specifically, how would such a model relate to the analysis of BR?
>
> *W2*
> - Out of the 36 datasets, only 6 were arbitrarily selected by us as popular UCI databases. The remaining 30 datasets were directly taken from the reference paper without any selection; all datasets from the reference paper were included.
> - All categorical variables were one-hot encoded. We did not observe any dependency between the number of binary features and the optimal BR value.
>
> *W3*
> The analyzed configurations (18 RF parametrizations and 10 BRs) were not selected individually for each dataset, but applied across all datasets. When training the classifier, each training instance consisted of features corresponding to the k_l values for a given dataset. Therefore, the number of training instances was equal to the number of datasets considered.

---

> > ### Comment · Reviewer_GY4H · 2024-11-24
> > **Response to author(s)**
> >
> > I would like to thank the author(s) for the response. However, I feel that the response does not directly address my key concerns. The experimental settings appear to be somewhat ad hoc and limited, making it difficult to draw general conclusions. Additionally, without further reasoning and insights, it is challenging to understand how the empirical findings derived from small classical datasets can be generalized to broader data and tasks. Therefore, I will maintain my original score.

---

### Comment · Area_Chair_953D · 2024-11-13
**authors - reviewers discussion open until November 26 at 11:59pm AoE**

Dear authors & reviewers,

The reviews for the paper should be now visible to both authors and reviewers. The discussion is open until November 26 at 11:59pm AoE.

Your AC

---

> ### Comment · Area_Chair_953D · 2024-11-25
>
> Dear reviewers,
>
> The authors have provided individual responses to your reviews. Can you acknowledge you have read them, and comment on them as necessary? The discussion will come to a close very soon now:
> - Nov 26: Last day for reviewers to ask questions to authors.
> - Nov 27: Last day for authors to respond to reviewers.
>
> Your AC

---

### Meta-Review · Area_Chair_953D · 2024-12-19

**Metareview:**

The paper provides an empirical study of Random Forests using a bootstrap sampling rate above 1. This contribution was considered insufficient by the reviewers, all of which recommended rejection.

**Additional Comments On Reviewer Discussion:**

N/A

---

### Decision · Program_Chairs · 2025-01-22

Reject